# Efficacy of cilastatin sodium in a translational large animal crush syndrome model
Adam C. Munhall [1], Mahaba B. Eiwaz[1], Jessica F. Hebert[1,2], Tahnee Groat[1], Nicole K. Andeen[3], Daniel Muruve[4], Ian J. Stewart [5], Lindsay Loss[6], Lydia Buzzard[6], Moqing Liu[6], Lylord Pierre[6], Louis Tinoco-Garcia[6], Samantha Durbin[6], Marissa Beiling[6], Karen Minoza [6], Joseph P. Garay [6], Martin A. Schreiber[6,7,9] & Michael P. Hutchens [1,2,8,9] ✉

## Abstract

**Background** Crush syndrome (consisting of hyperkalemia, acidosis, hypocalcemia, and acute kidney injury), is the second-most common cause of death in earthquakes, and a frequent cause of critical illness after burn, blast, and prolonged immobility. No specific treatment exists; supportive treatment is burdensome, contributing to deaths in austere environments, especially disasters and conflicts. There is urgent need for specific treatment which reduces burden of care. Crush syndrome is dependent on the renal megalin-dependent endocytic system. We investigated whether cilastatin sodium, a megalin inhibitor which is US Food and Drug Administration-approved for another purpose, has efficacy as a crush syndrome treatment in a highly translational large animal trauma model.
**Methods** Anesthetized 40 kg female pigs received blunt muscle injury and 48 h protocolized critical care management. Cilastatin sodium or vehicle was administered 30 minutes after injury in randomized, blinded fashion. Renal function and injury were assessed by repeated quantification of iohexol clearance, serial plasma assessment, and histopathologic analysis. Linear mixed models and Kaplan-Meier analysis were used to assess differences. A power estimate for a clinical trial was performed.
**Results** Here we show that cilastatin has efficacy to reduce kidney impairment from crush syndrome, resulting in increased measured glomerular filtration rate and reduced creatinine, histopathologic kidney damage, and need for treatment of hyperkalemia. Animals receiving cilastatin excrete more myoglobin in the urine and are more likely to recover from acute kidney injury. The effect size suggests feasibility of future clinical trials.
**Conclusions** Cilastatin sodium has efficacy to ameliorate crush syndrome in a translational large animal model These results support further efforts to translate this potential therapy.

## Plain language summary

Crush syndrome occurs when people are crushed under rubble, or by explosions. It is the second-most common cause of death in earthquakes, and frequent in war. Crushing muscle releases a toxin which causes kidney failure and death from excessive blood potassium, which stops the heart. Treatment requires intensive care, which is often not available in disasters and war zones. Using anesthetized pigs with experimental crush syndrome, we found that the drug cilastatin reduced kidney failure and excessive blood potassium, and increased the likelihood of kidney recovery. Because cilastatin is already approved for another purpose, the path to use for treatment is less challenging than for new drugs. These results may pave the way for human trials of a new treatment.

Crush syndrome (CS) is potentially lethal hyperkalemia, acidosis, hypocalcemia, and acute kidney injury (AKI) caused by physical destruction of skeletal muscle. Muscle destruction (rhabdomyolysis) can result from crush (as in building collapse), blast shockwave, or severe muscle swelling (from ischemia or overuse)[1–11]. Death and severe illness result from the interaction of skeletal muscle myoglobin with renal proximal tubules[12–14]. Myoglobin-induced kidney injury has several mechanisms, including direct induction of ferroptosis, vasoconstriction, and tubular obstruction[15–17]. However, our work in mice demonstrated that loss of glomerular filtration function, which leads to the lethal consequences of CS, is dependent on tubular cell uptake

[1]Department of Anesthesiology & Perioperative Medicine, Oregon Health & Science University, Portland, OR, USA. [2]Divison of Nephrology & Hypertension, Oregon Health & Science University, Portland, OR, USA. [3]Department of Pathology, Oregon Health & Science University, Portland, OR, USA. [4]Division of Nephrology, Cumming School of Medicine, University of Calgary, Calgary, Alberta, Canada. [5]Division of Nephrology, Hebert School of Medicine Uniformed Services University, Bethesda, MD, USA. [6]Trunkey Center for Civilian and Combat Trauma, Oregon Health & Science University, Portland, OR, USA. [7]Department of Surgery, Hebert School of Medicine, Uniformed Services University, Bethesda, MD, USA. [8]Operative Care Division, Portland Veterans Administration Medical Center, Portland, OR, USA. [9]These authors contributed equally: Martin A. Schreiber, Michael P. Hutchens. ✉e-mail: hutchenm@ohsu.edu

via the megalin-mediated endocytic system[18]. Treatment is supportive, consisting of early intravenous fluid (to dilute myoglobin in the renal tubule), critical care, and potentially hemodialysis; together, these treatments are burdensome to providers and medical systems. CS commonly occurs in clusters due to disasters/conflicts[19,20]; these are environments in which advanced medical care (especially dialysis) is limited, increasing mortality[19,21–29]. Therefore, there is an urgent need for specific treatment which reduces the burden of care.

Cilastatin sodium is US Food and Drug Administration-approved in combination with the antibiotic imipenem; it was identified as an inhibitor of renal dipeptidase (DHP-1) in the 1980s, and more recently as an inhibitor of megalin-mediated endocytosis[30]. Cilastatin's renoprotective action in diverse rodent AKI models[31–35] is partly attributed to DHP-1 inhibition. It is also renoprotective in rhabdomyolysis-induced AKI by a megalin-dependent mechanism[18]. These data support the repurposing of cilastatin sodium for the treatment of CS. While mouse models have established critical mechanisms and physiology, a large animal study in a clinically-relevant model is necessary to determine the translatability of rodent discoveries prior to human investigation. *Sus scrofa* have similar size, pharmacology, and renal physiology to humans and a long accepted role in surgical discovery translation, including for traumatic injury[36–38]. Here, we determine the efficacy of cilastatin sodium in a clinically-relevant, pig CS model.

We find that cilastatin sodium administration 30 min after injury reduces kidney functional and histopathologic damage due to CS. Additionally, cilastatin sodium reduces the need for treatment to limit life-threatening hyperkalemia and increases the rate and likelihood of rapid recovery.

## Methods
**Detailed methods are provided in supplementary methods**
**Animals, randomization, and animal groups.** Animals used for these investigations were Female Yorkshire-Landrace crossbred pigs weighing 40–50 kg. Only female pigs were used because of tractability and because prior studies in mice used only male mice, and efficacy in female animals was unknown. All procedures were approved by the Institutional Animal Care and Use Committee of Oregon Health & Science University and the Animal Care and Use Review Office of the US Army Medical Research and Development Command. Animals were block randomized in 2:2:2:1 cilastatin:cilastatin+calcitriol:vehicle:no impact ratio. Animal group size was determined by a priori power analysis performed using effect size data from a prior study conducted in mice[18]. Animal surgeons and the renal pathologist were blinded to drug treatment; the renal pathologist was also blinded to the surgical model (no-impact vs. impact). In a preplanned analysis, there was no difference between the cilastatin and cilastatin+calcitriol groups. Group assignment and experimental numbers are reported in Fig. 1.

**Drug activity assurance and preparation.** To assure equivalent activity across batches, aliquots of cilastatin sodium were subjected to renal dipeptidase activity assay[39,40]. To prepare drug bags, on the morning of surgery, animal weight was measured. Cilastatin sodium was prepared by adding 100 mg/kg of previously-prepared concentrated solution (Cilastatin sodium 400 mg/mL in Plasma-Lyte A) to a 500 mL bag of Plasma-Lyte A. Vehicle bags contained an equal volume of Plasma-Lyte A and were visually identical to cilastatin bags. Calcitriol (1 µg) was added to a second, 125 mL bag; an additional calcitriol vehicle bag was administered to animals not receiving calcitriol.

**Large animal crush syndrome model.** After induction of general anesthesia, tracheostomy, and urinary, arterial and venous catheter placements were performed. Anesthesia was maintained throughout the experiment with isoflurane (1–3%) and infusion of ketamine (5 mg/kg/h). Electrocardiography, end-tidal carbon dioxide, arterial blood pressure, temperature, and oxygen saturation were continuously monitored.

Hypotension (MAP < 40 mmHg for >19 min) was treated with intravenous fluids (protocol, Supplementary Methods). The bilateral thigh muscles were padded with cardboard and then impacted twice each (total 4 impacts) with an explosive-driven captive bolt (Schermer, Padeborn, Germany) driven by 200 g explosive propellant (Schermer, Padeborn, Germany). After impact, intravenous lactated Ringer's solution with 5% dextrose was initiated (10 mL/kg/hr for the first 3 h, 5 mL/kg/hr for the following 3 h, and then 2.5 mL/kg/hr for 40 h). Thirty minutes after impact, at study time zero, the drug or vehicle was administered intravenously over 5 min. In addition to a baseline draw after catheter placement, blood was drawn for analysis (iStat, Abbott, Orlando, FL) immediately before drug administration and at 2, 6, 12, 18, 24, and 48 h after drug administration; urine samples were collected simultaneously. Blood potassium >5.0 was immediately treated with insulin, calcium gluconate, and dextrose (protocol, Supplementary Methods). 48 h after drug administration, animals were euthanized by pentobarbital overdose. Event sequencing in the model is presented in schematic form in Fig. 2.

**Measurements performed on blood and urine samples.** Sodium, potassium, chloride, bicarbonate, creatinine, urea nitrogen, ionized calcium, prothrombin time, pH, total carbon dioxide, partial pressure of carbon dioxide, partial pressure of oxygen, oxygen saturation, and base excess in blood samples were determined using an iStat (Abbott, Orlando, FL) autoanalyzer. Creatine kinase was measured in plasma samples by the OHSU clinical laboratory using a BeckmanCoulter AU analyzer. Urine and plasma myoglobin were measured by ELISA using a commercially-available kit (MYO-9, Life Diagnostics, West Chester, PA) following manufacturer instructions. Urine total porphyrins were measured after diluting in oxalic acid and heating[41]. To determine the acute effects of cilastatin on non-renal targets, systemic physiology was assessed during the 12 h after administration. This time limit was chosen to represent a conservative estimate of the maximal plasma residence time for cilastatin sodium, as the plasma half-time in estimated as 45–60 min[42,43] and 12 h accounts for >5 half times even in the presence of >50% GFR loss.

**Quantification of kidney function, including measurement of glomerular filtration rate (GFR).** Five milliliters of intravenous iohexol (OmniPaque 300, GE Healthcare) was administered intravenously at 1, 19, and 43 h. Blood samples (~2 mL each) were collected 15, 30, 90, 120, 180, 240, and 300 min after iohexol injection. Liquid chromatography/mass spectrometry/mass spectrometry (LC/MS/MS) was used to determine the iohexol concentration in each sample. Concentration-time data were then fit to a 2-compartment model using the R package *stats*. If the fit for the 2-compartment model was poor, a 1-compartment model was fit. GFR was calculated as (iohexol dose)/(area under the fitted curve)[44–48]

**Pathologic assessment of kidney injury.** Qualitative and quantitative assessment of kidney injury was performed by a blinded renal pathologist (NCA) on hematoxylin and eosin-stained, formalin-fixed, paraffin-embedded kidney slices. The tubular damage score was determined from 16 random high-power fields per animal as described[14].

**Assessment of nonrecovery from AKI and clinical trial simulation.** Increasing preclinical and clinical data suggest that preclinically-defined renal repair, maladaptive repair, and AKI-CKD transition are likely associated with clinical AKI recovery trajectories, including rapid recovery, delayed recovery, and nonrecovery (Reviewed in Ostermann et al.[49]). These trajectories are also clinically associated with mortality and remote organ dysfunction. To assess this important translational outcome, we identified relevant thresholds from the clinical literature for rapid recovery at 48 h and applied them. Consensus guidelines define persistent AKI as that which continues beyond 48 h[50]. Applicable (within 48 h) clinical thresholds for rapid recovery, which reduced risk of dialysis, mortality, or development of chronic kidney disease, were identified as 30%

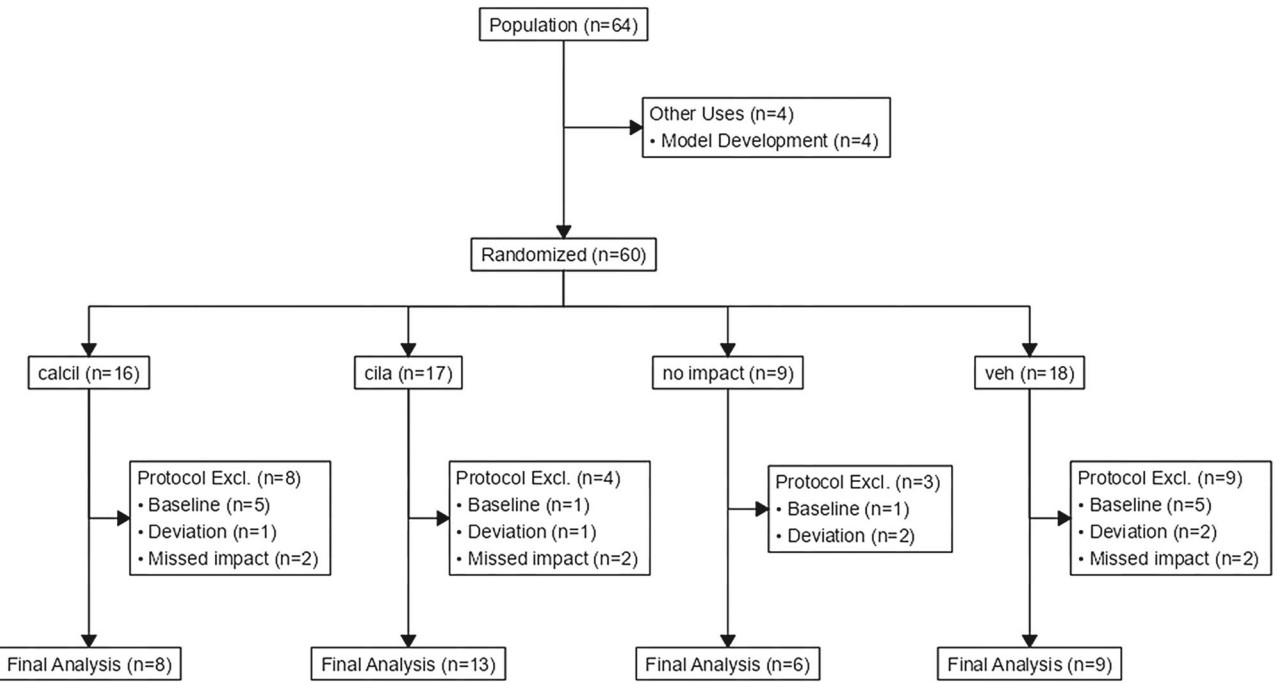

**Fig. 1 | CONSORT diagram depicting allocation of animals.**

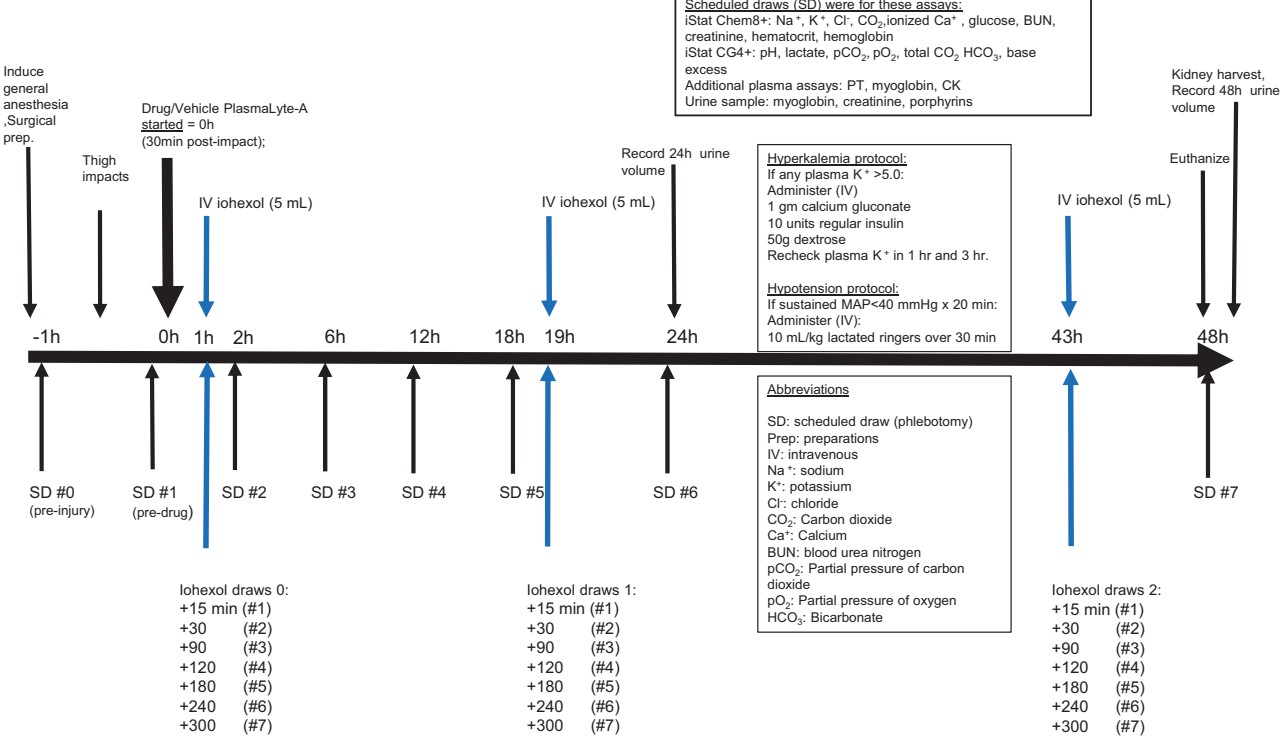

**Fig. 2 | Schematic of the 48 h animal surgical model, interventions and assessments.** After anesthesia induction and surgical preparation, impacts were delivered, and 30 min later (at time zero) drug or vehicle was administered. Animals then underwent a total of 48 h of general anesthesia with continuous physiologic monitoring and close blood chemistry surveillance, including measurement of glomerular filtration rate (by iohexol clearance). Animals were euthanized at 48 h.

recovery of creatinine[51] and failure to recover to 70% of baseline eGFR[52,53]. To enable powering for a future clinical trial from this 48 h study, we performed a Monte Carlo simulation using the categorical variable of recovery to 70% of the maximal creatinine by 48 h as the outcome.

**Statistics and reproducibility.** Statistical analysis was performed using the R statistical language. All statistical analyses performed and necessary to replicate the results are documented in the shared code (See "Data availability" and "Code availability"). Comparisons between groups were conducted using mixed models linear regression, and residuals evaluated

for concordance with the assumptions of distribution using residual vs. predicted value and quantile plots. Except as noted, results are presented in text and figures as mean ± standard error of the mean (SEM). In several cases, outliers resulted in violation of assumptions of distribution; in these cases, robust linear mixed models were instead employed, and results are presented as median (25th percentile, 75th percentile). Because of seasonal and sequence-based variation in baseline variables (Supplementary Results), model terms included the sequential animal ID as a random factor. Cilastatin and cilastatin + calcitriol groups were pooled for analysis after preplanned analysis (Supplementary Results) demonstrated no difference in pre- or post-drug physiologic measurements, and no difference in outcome. Adjustment for multiplicity and repeated measurements was performed using the Šidák correction. Analysis of survival and time-to-intervention was performed using Kaplan–Meier analysis and log-rank tests of significance. Analysis of categorical outcomes was performed with $X^2$. Replicates documented in this manuscript are experimental replicates (e.g., each replicate is the result of an independent experiment). Overall sample size is documented in Fig. 1, and specific sample size information for each depicted data measure is provided in figure captions.

## Results

### Impact injury model

All randomized animals underwent successful induction of general anesthesia. Seven animals experienced significant protocol deviation and/or death before 48 h, primarily during the instrumentation procedure. Thirteen animals demonstrated severe physiologic abnormalities on pre-injury baseline assessment (creatinine, temperature, plasma myoglobin, or potassium >2 SD from the mean). Delivery of explosive-driven impacts caused CS in 45/51 (88%) animals assigned to receive impact. In 6 animals (3 vehicle-, 3 cilastatin-treated), despite apparent impact, CK was not elevated by 6 h (baseline to 6 h change in CK: 27 ± 10 IU/kg vs. 61 ± 4 IU/kg, $p = 0.0012$). Based on these a priori exclusion criteria, these animals were excluded from further analysis (Supplementary Data: Exclusions). Cilastatin batches had identical dipeptidase-1 activity and were not different from comparator preparations used in our small animal studies[18] (Supplementary Fig. 1). Protocolized treatment resulted in administration of 138.2 ± 6.2, 147.8 ± 5.2, and 136.0 ± 3.2 mL/kg of intravenous fluid over 48 h, respectively to no impact, vehicle-, and cilastatin-treated animals ($p > 0.05$). Overall survival to the 48 h endpoint was 89%, also not affected by treatment ($p = 0.86$, Supplementary Fig. 2).

### Impact injury model caused crush syndrome and critical illness

Bilateral thigh impact caused rhabdomyolysis (creatine kinase, CK 4400 ± 1420 at 6 h and 16740 ± 1480 IU/L at 48 h, Fig. 3A) and systemic manifestations of CS, including hyperkalemia and hypocalcemia. Due to prolonged immobility under general anesthesia, even nonimpacted animals developed elevated plasma CK acute kidney injury (oliguria, and reduced GFR), but impact-specific findings were accelerated by impact such that CK in impacted animals was more than double that of non-impacted animals at all time points after 6 h ($p < 0.05$ at 6 h and <0.001 at >6 h). All impacted animals developed hyperkalemia requiring treatment before 6 h, while no non-impacted animals did (3B, $p < 0.05$ by log-rank test). Accordingly, animals subjected to impact required 1.4× as many interventions for hyperkalemia over 48 h (3 C, 7 (4, 8) vs. 10 (8, 12) at 48 h, $p = 0.0029$). Hypocalcemia (a hallmark of CS) developed in impact-treated animals, but did not develop in non-impact animals (3D, 1.3 ± 0.02 vs. 1.23 ± 0.02 at 18 h, $p = 0.0490$). Blood pressure was similar, although trending lower in animals subjected to impact, and this was not reflected in heart rate or total intravenous fluids administered(3E, Supplementary Fig. 3A, B). Oliguria developed rapidly in both groups (3 F) and was reflected in dramatically increased tubular damage score (3 G) and reduced GFR (3H). Plasma lactate and creatinine were not significantly altered by impact (Supplementary Fig. 3C, D). We conclude that the thigh impact injury model caused CS.

### Acute and off-target effects of cilastatin administration

To determine whether cilastatin administration acutely and directly altered non-renal physiology, we assessed physiologic measures of homeostasis relevant to CS after randomization and drug or vehicle administration in impacted animals. Up to 12 h after treatment (more than 10× cilastatin's expected plasma half-time), there was no difference between cilastatin and vehicle in mean arterial pressure, heart rate, temperature, administered intravenous fluid, prothrombin time, plasma ionized calcium, and base excess (Fig. 4A–H). However, after 12 h, mean arterial pressure (49.67 ± 1.5 vs. 44.07 ± 2.25 mmHg, $p = 0.0408$ at hour 32), ionized calcium (1.29 ± 0.01 vs. 1.23 ± 0.02, mmol/L, $p = 0.0129$ at 18 h), and base excess (all times ≥12 h, Fig. 4H) were higher, and heart rate lower (from 12–24 to 31–34 h, Fig. 4B), in cilastatin-treated animals. Urine output was 67% greater during the first hour after cilastatin administration than after vehicle treatment (Fig. 4E, 4.89 ± 0.14 vs. 2.92 ± 0.21 mL/kg/hr, $p < 0.0001$), and similar at all other times. Since 12 h is >10× the half time of cilastatin and 12 h coincides with the onset of hyperkalemia, severe AKI, and elevation of plasma CK, we conclude that cilastatin administration did not directly alter mean arterial pressure, heart rate, temperature, prothrombin time, plasma ionized calcium, or base excess. A transient increase in urine output was observed, potentially reflecting a diuretic effect.

### Effect of cilastatin sodium on AKI and recovery

In a preplanned analysis (Supplementary Analysis 1), we determined that the addition of calcitriol to cilastatin had no physiologically significant effect, and therefore, in further analyses, we pooled results from cilastatin-treated and cilastatin+calcitriol-treated animals. Cilastatin administration resulted in increased GFR 48 h after injury (Fig. 5A, 96.16 ± 5.43 vs. 73.23 ± 8.36 mL/min, $p = 0.024$), the preplanned primary outcome of the study. Plasma creatinine was reduced in cilastatin-treated animals 24 h after injury (Fig. 5B, 2.6 (2.2, 3.7) vs. 3.5 (2.7, 4.2), $p = 0.041$), and urea nitrogen was reduced by cilastatin treatment 18 h after injury (Fig. 5C). On pathologic assessment, all impacted animals demonstrated tubular injury, including protein-containing tubular casts, with some also demonstrating sparse obstructive debris-containing casts. There was no apparent treatment-related difference in location or quantity of either type of tubular cast. Quantitative pathologic assessment (Fig. 5D, additional images in Supplementary Fig. 4) demonstrated that cilastatin treatment conferred 45 ± 33% reduction in tubular damage score ($p = 0.029$). Because temporal trends of creatinine and GFR suggested a role for cilastatin in recovery from AKI, we investigated the change in GFR with time in individual animals. The rate of change in GFR from 24 to 48 h was increased by cilastatin (Fig. 5E, interaction $p = 0.0193$), further supporting its role in reversal of AKI. Since recent clinical studies and expert guidelines have identified reversal of AKI within 48–72 h ("rapid reversal") as reducing risk of mortality, requirement for dialysis, and development of chronic kidney disease[50,53–55] (Reviewed in Ostermann et al.[49]), we assessed whether rapid reversal of AKI within 48 h depended on treatment. Recovery to <70 of the maximal creatinine occurred in 57.1% of cilastatin-treated animals and 11.1% of vehicle-treated animals ($p = 0.041$). Time-to-event analysis confirmed that recovery to <70% of maximal creatinine was greater in cilastatin-treated animals (Supplementary Fig. 5, $p = 0.02$). Accordingly, 94.4% of cilastatin-treated animals demonstrated normal measured GFR[46] at 48 h while 50% of vehicle-treated animals did ($p = 0.0193$). We conclude that cilastatin sodium administration 30 min after blunt injury resulted in amelioration of crush-induced AKI, and a more rapid and greater likelihood of recovery of renal function by 48 h as reflected by measured GFR and creatinine.

### Effect of cilastatin sodium on hyperkalemia

Next, we determined whether cilastatin treatment mediated the lethal complication of CS, hyperkalemia. 38/39 (97%) animals developed serum potassium >5.0 mmol/L, which was the protocolized threshold for treatment. Therefore, we assessed the cumulative number of hyperkalemia

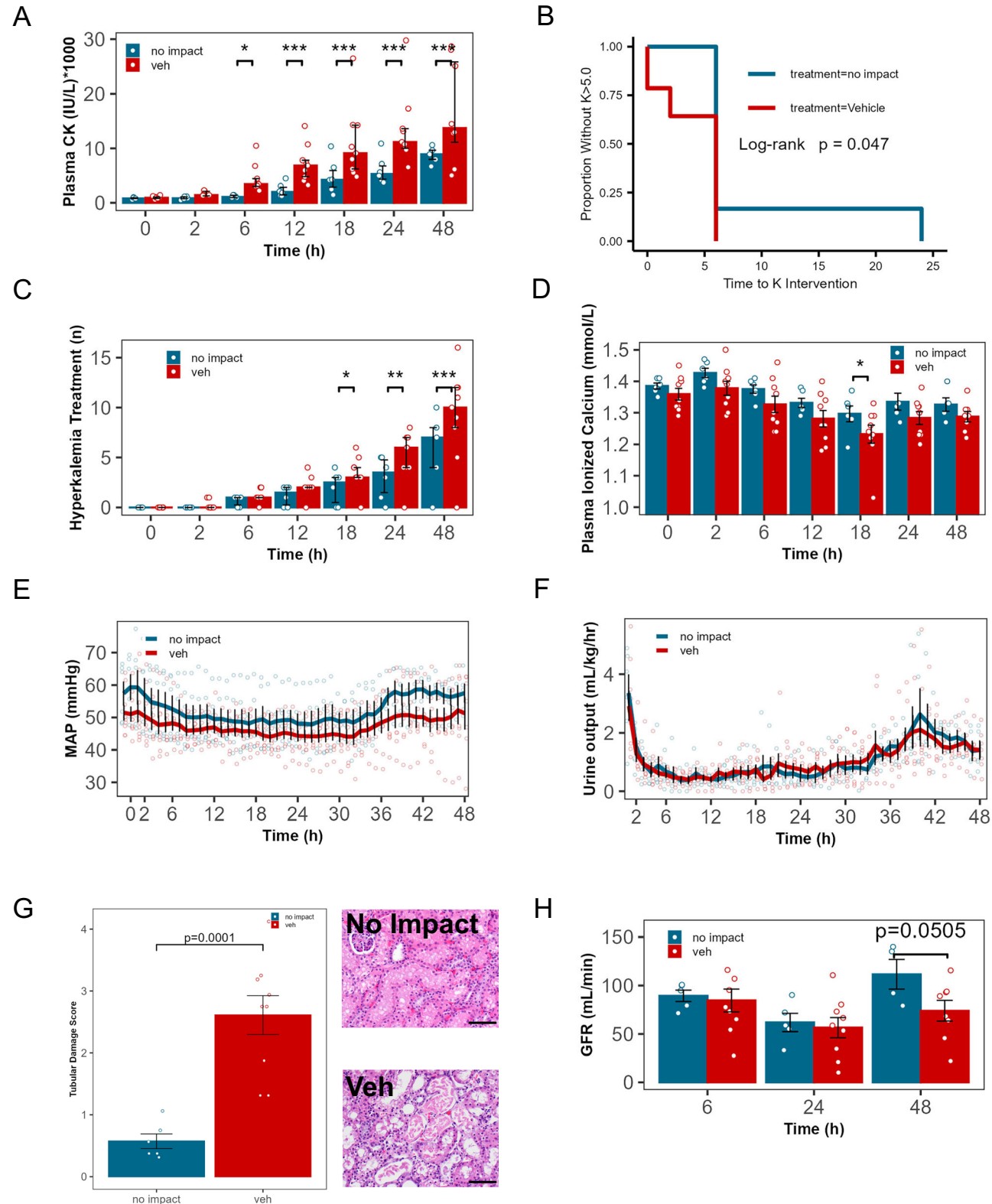

protocol activations and the time to event for hyperkalemia protocol activation. Cilastatin reduced the cumulative number of hyperkalemia interventions required (10 (8, 12) in veh vs. 4 (1, 11) in cil, $p = 0.0005$, Fig. 5F), and accordingly, the time to first treatment ($p = 0.011$, Fig. 5G). Therefore, cilastatin administration reduced the need to treat for hyperkalemia due to CS.

**Cilastatin sodium effect on renal myoglobin excretion**

Because a prior study indicated increased clearance of myoglobin in cilastatin-treated mice[18] we assessed myoglobin urinary excretion. These results are illustrated in Fig. 6. Cilastatin did not alter plasma CK overall or at any time point (Fig. 6A). Because CK is too large for renal filtration, we interpret this to mean that muscle injury and myoglobin release were not

**Fig. 3 | A crush injury model in 40 kg swine causes traumatic injury, critical illness, and rhabdomyolysis-induced acute kidney injury.** Compared with vehicle-treated animals which received impact injury (veh, $n = 9$), animals which did not receive impact (but were identically treated with 48 h of general anesthesia and protocolized critical care including fluid resuscitation, $n = 6$) demonstrate delayed and half-magnitude rise in creatine kinase (CK, $p = 0.039$, 0.0005, 0.0002 0.0001, <0.0001, respectively, at 6, 12, 18, 24, and 48 h). The rise in CK in no impact animals was likely due to prolonged immobility. No impact animals required intervention for hyperkalemia later (**B**) and received fewer total interventions for hyperkalemia (**C**, $p = 0.0451$, 0.0030, <0.0001, respectively at 18, 24, and 48 h). Plasma ionized calcium (**D**) was lower in impacted vs. non-impacted animals at 18 h, consistent with crush-induced reduction of plasma calcium ($p = 0.0490$). Mean arterial pressure ($p = 0.062$) and urine output were not different between groups. (**E**, **F**). Impact greatly increased pathologist-determined tubular damage score (**G**, hematoxylin and eosin-stained representative image, scale bar: 50 μm), but did not significantly reduce glomerular filtration rate at any time point (**H**) ($n = 6$ no impact,9 veh, *=$p < 0.05$, **=$p < 0.01$, **A** robust linear mixed models regression, **C**–**F**, **H** linear mixed models regression **B** log-rank test, **G** Welch test. **A**, **C** error bars: median ± interquartile range, **D**–**H** error bars: mean ± SEM.) Scale bars: 50 μm.

changed by cilastatin administration. Nonetheless, cilastatin reduced plasma myoglobin in time-dependent fashion ($p$ for interaction =0.024); this was >40% reduced at 12 h and >60% reduced by 24 h (Fig. 6B, 2499 (1869, 2970) vs. 1339 (1011, 2549) in cilastatin-treated animals, $p = 0.005$ at 12 h, and 4181 (2530, 6352) vs. 1425 (1002, 2387) ng/mL, $p < 0.0001$, at 24 h). Urine total protein excretion (as a fraction of creatinine excretion) was increased by cilastatin administration (Supplementary Fig. 6A). Urine myoglobin measurements were lower and less consistent than previously measured in mouse models[18], likely due to (well-described) large-mammal urine matrix effects[56,57]. Nonetheless, fractional excretion of myoglobin (Fig. 6C) and urine myoglobin (Supplementary Fig. 6B) were increased by cilastatin at 2 h, although total 48 h myoglobin excretion was not increased by cilastatin (Supplementary Fig. 6C, mechanism summarized in Supplementary Fig. 6D). Since myoglobin contains porphyrin moieties, to corroborate urine myoglobin, we quantified total urine porphyrins. Accordingly, 48 h total porphyrin excretion was increased by cilastatin (Fig. 6D 52.6 ± 31.3 in veh vs. 110.4 ± 96.4 μg porphyrin/μg creatinine*48 h in cilastatin-treated animals, $p = 0.02$). We conclude that cilastatin did not alter plasma CK but increased excretion of the toxic, iron-containing moiety of myoglobin.

## Power estimate for a clinical trial

To the best of our ability to determine, no clinical data exists with which to power a clinical trial of a potential therapeutic in CS patients. Since this step is necessary for translation, we performed power estimation using a Monte Carlo simulation. Since rapid recovery from AKI strongly associates with reduced risk of acute and chronic complications, which could support patient-centered outcomes[58], we selected recovery from maximal creatinine by 48 h and performed a Monte Carlo simulation to estimate the necessary experimental number for a randomized clinical trial of cilastatin sodium. Based on the effect observed, for a stringently-designed study with an α error of 1% and β of 0.9, a minimum of 124 patients in both groups would be required. Figure 7 illustrates the results of the simulation.

## Discussion

Specific treatment for CS could greatly increase access to treatment in disasters and conflicts, likely improving outcomes. Our most important finding is that cilastatin sodium has efficacy to reduce acute kidney injury due to traumatic crush injury in a clinically relevant large animal model. We induced crush injury in 40 kg pigs with explosive-driven blunt force and provided 48 h of protocolized critical care based on clinical guidelines[6,59]; kidney function was assessed with iohexol clearance, a gold-standard method, histopathology, and guideline-recommended measures. Blunt impact caused CS, including CK elevation to levels consistent with human CS[60], acute tubular damage, severe hyperkalemia, and hypocalcemia. Randomized, blinded administration of cilastatin sodium reduced kidney functional loss and tubular damage, demonstrated by increased GFR, attenuated creatinine, and almost 50% reduction in renal tubular damage score. These findings recapitulate our prior study in mice[18]. Although no clinical studies of specific drug treatment for CS have been reported, pre-clinical studies (reviewed in Hebert et al.[61]), suggest a number of strategies, including anti-inflammatory drugs[62–64], antioxidants[65–67], and mesenchymal stem cell preparations[68,69], may have efficacy, demonstrating that

mechanistic approaches may yield successful translation. A salient feature of cilastatin sodium is that it is a repurposable drug with an extensive safety record, now with demonstrated efficacy in large animals using human-relevant outcomes and treatment similar to human standard-of-care. Therefore, in order to facilitate translation, we also assessed whether cilastatin administration influenced renal recovery or the need for treatment for lethal hyperkalemia. Cilastatin increased the rate and likelihood of renal recovery and reduced the need for treatment of potentially lethal hyperkalemia. Failure to rapidly recover renal function after AKI strongly associates with the onset of chronic kidney disease and the need for dialysis[50,52], as well as worsening and new-onset cardiovascular disease and mortality[51,70,71]. Similarly, CS-induced hyperkalemia is frequently lethal and often requires dialysis, which can be inaccessible, particularly in disasters such as earthquakes[5,19,23]. Therefore, the increased likelihood and rate of renal recovery, and reduced necessity for hyperkalemia treatment in cilastatin-treated pigs are important outcomes that may suggest potential metrics for clinical trial design as part of additional translational research.

Cilastatin has an extensive safety record and very low toxicity[42]. The 100 mg/kg we delivered is a higher dose than currently FDA-approved in combination with imipenem (500 mg every 6 h, or ~30 mg/kg over 24 h). Therefore, a secondary objective was to seek out potential adverse effects of cilastatin in critically ill animals. Heart rate, temperature, mean arterial pressure, pH, base excess, ionized calcium, and prothrombin time were all unaltered by cilastatin for 12 h after administration, suggesting no measured acute toxic effect of cilastatin. Surprisingly, we found that urine output was increased in the hour after cilastatin administration. Despite an extensive literature review, no prior study assessing 1 h urine output after any dose of cilastatin could be found. Therefore, this observation requires additional study. Effects observed after 12 h were all salutary and consistent with amelioration of CS, rather than direct effects of the drug. These results support the safety of 100 mg/kg cilastatin sodium in CS, adding to extensive toxicological data supporting clinical use of cilastatin[42,43,72,73]. A phase I trial (NCT03595189) including dosing similar to that used in our study was completed in 2018. Results of that study and perhaps additional pharmacokinetic/pharmacodynamic study will therefore be required as part of translational research.

In mice with glycerol-induced rhabdomyolysis, cilastatin increased urinary clearance of myoglobin tenfold; this effect, and amelioration of kidney injury, was dependent on renal megalin, suggesting that urinary myoglobin clearance was an important mechanism of protection. To confirm the mechanism, we assessed urine and plasma myoglobin. Urine myoglobin measurements, despite considerable effort, produced inconsistent and apparently insensitive results, likely because of large animal urine matrix effects[56,57]. However, assessment of fractional excretion of myoglobin and its constituent porphyrins, as well as plasma myoglobin, confirmed that urine excretion of myoglobin was increased by cilastatin, perhaps not to the degree observed in mice. We speculate that measurement of repeated spot urines in pigs, rather than true 24 h samples as performed in mice, may account for this difference. However, it should be noted that cilastatin has demonstrated additional renoprotective properties[31–35,74–80] beyond increasing myoglobin clearance in CS, and it is possible these effects, believed to be exerted through interaction with renal dipeptidase, also

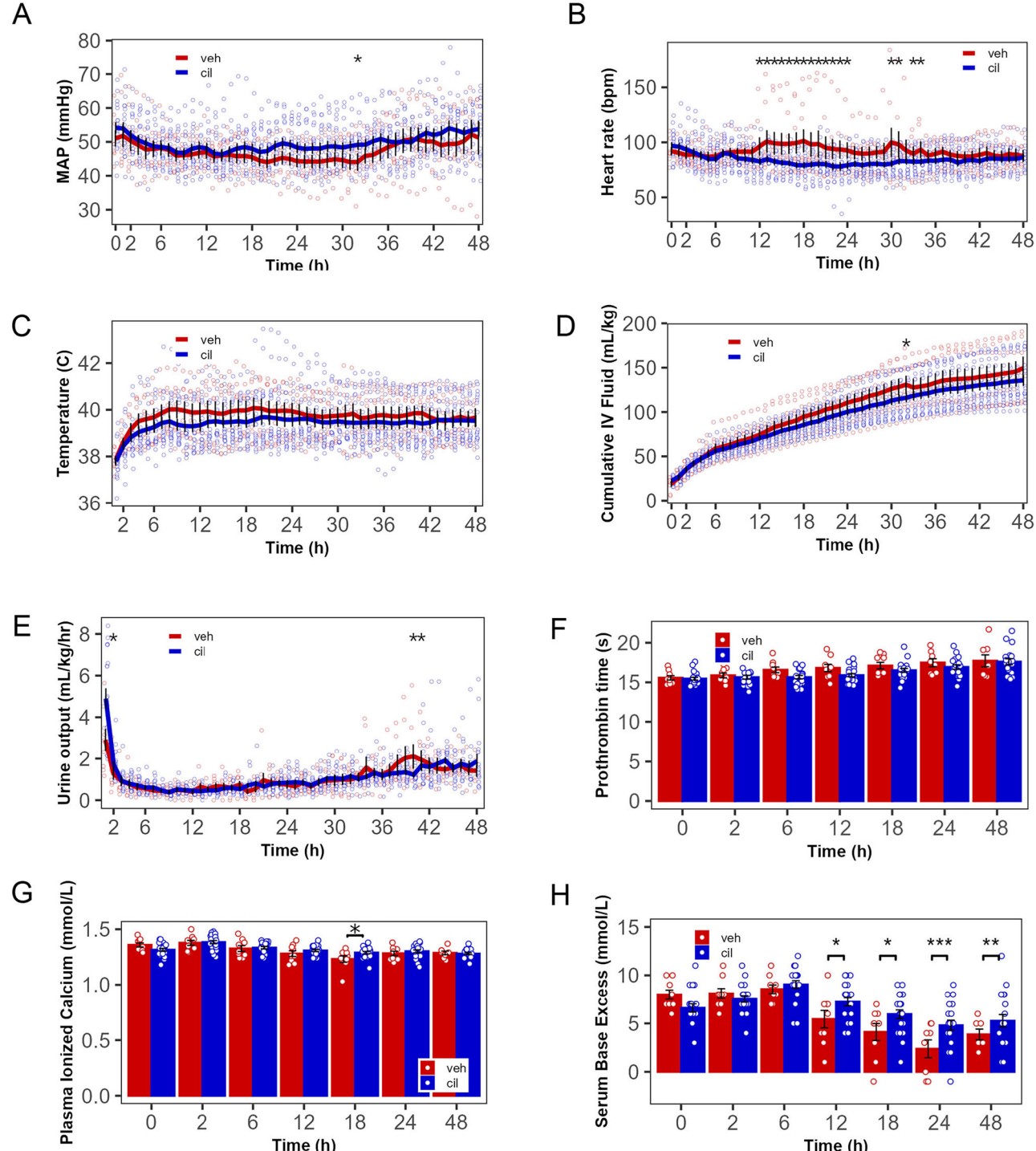

**Fig. 4 | Cilastatin sodium (cil) administration had no acute direct ( ≤ 12 h) effect, and minimal delayed effect on cardiovascular or other systemic physiology (relative to vehicle treatment, veh) at the time of administration. A** Mean arterial pressure (MAP) was overall not significantly altered, although at 33 h, MAP was lower in Cil-treated animals. **B** Heart rate was also not overall significantly altered (mixed-model $p$ value > 0.05), although heart rate was reduced at multiple individual time points after 12 h. **C** Cil administration did not alter temperature or (**D**) total fluid intake (with the exception of a single time point at 32 h. **E** Urine output was higher in cil-treated animals and was 60% higher in the first hour after Cil administration. **F** Prothrombin time was not altered. **G** Plasma ionized calcium was increased by cil at 18 h ($p = 0.0129$), the same time point at which ionized calcium was decreased by impact compared with no impact (Fig. 3). **H** Base excess, although not changed prior to 12 h, was significantly increased at all time points starting 12 h after cil administration ($p = 0.0402, 0.0353, 0.0007, 0.0059$, respectively, for 12, 18, 24, and 48 h. $n = 9$ veh, 21 cil, *=adjusted $p < 0.05$, **=adjusted $p < 0.01$, ***=adjusted $p < 0.001$, **A–H**: linear mixed models regression. **A–H** error bars: mean ± SEM.)

contributed to the improved renal function that resulted from cilastatin administration.

Our study has limitations. The study was designed for 48 h survival, and therefore injury was calibrated to ensure adequate 48 h survival; more severe injury may have additional effects not assessed in this work.

Additionally, although AKI was resolving at 48 h and cilastatin increased the number of animals with improvement in renal function at that time point, longer-term outcomes of CS or effects of cilastatin beyond 48 h were not assessed, and the reduced risk of CKD cannot be inferred from this data. Longer-term studies should be performed to evaluate

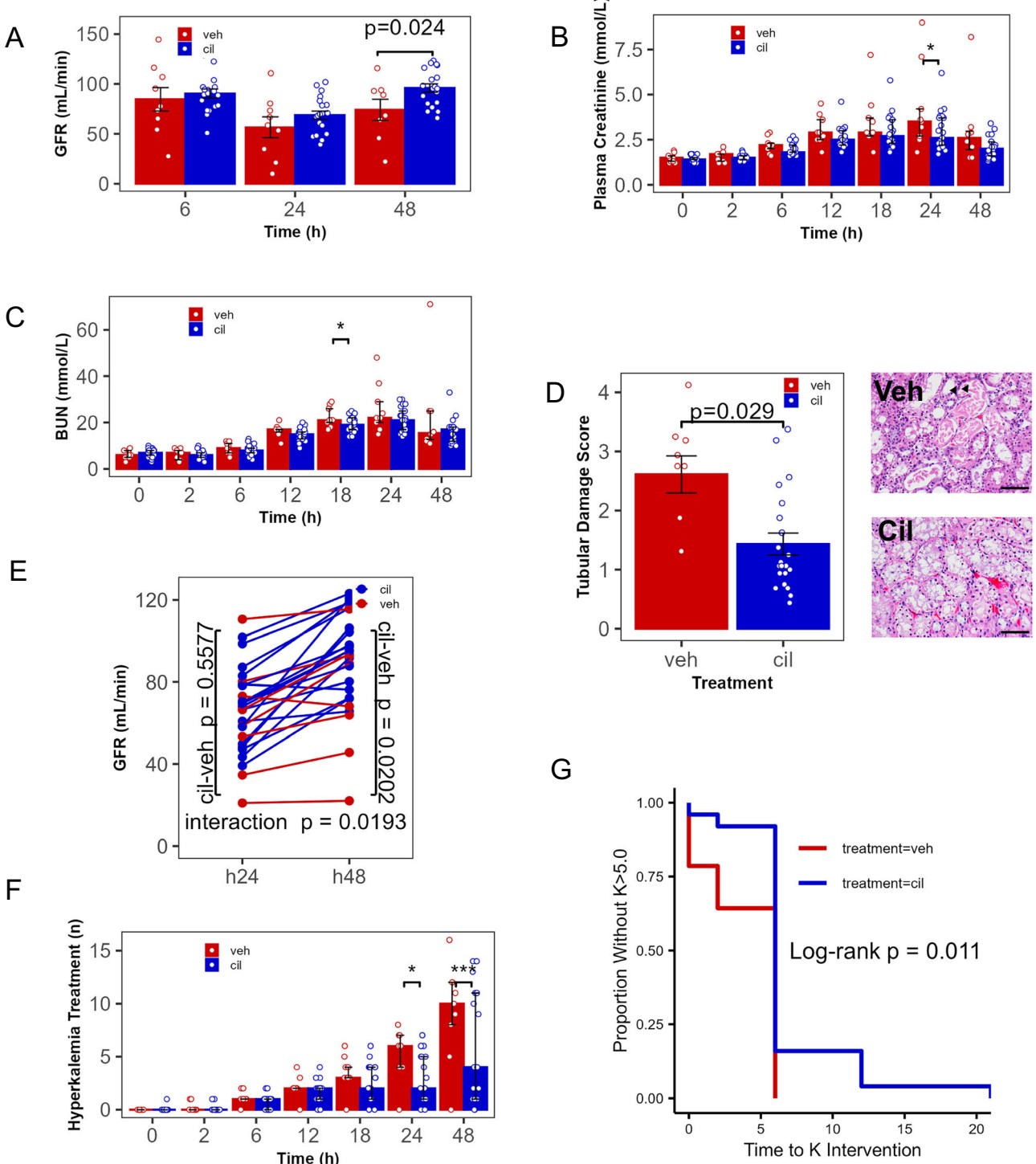

**Fig. 5 | Cilastatin sodium ameliorated crush-induced AKI. A** Glomerular filtration rate (GFR) was higher 24 h ($p = 0.0409$) after drug administration in animals treated with cilastatin (cil) compared with vehicle (veh). **B** Plasma creatinine was reduced by cilastatin administration. **C** Blood urea nitrogen (BUN) was reduced 18 h after cilastatin administration ($p = 0.0245$). **D** Cilastatin administration resulted in 45% reduction in tubular damage score, primarily driven by reduced tubular necrosis (arrowheads, necrotic debris, and necrotic tubular cell, hematoxylin and eosin-stained representative image, scale bar: 50 μm). **E** Cilastatin administration increased the rate of GFR recovery from 24 to 48 h. **F** Cilastatin administration reduced the cumulative number of protocolized interventions required for hyperkalemia (plasma $K > 5.0$) and delayed the onset of hyperkalemia (**G**, $p = 0.0003$ and <0.0001, respectively at 24 and 48 h. $n = 9$ veh, 21 cil, *$=p < 0.05$, **$=p < 0.01$, **A, E, F**: linear mixed models regression **B, C**: robust linear mixed models regression, **D**: Welch test **G**: log-rank test, **A, D** error bars:mean ± SEM), **B, C, F** error bars: median ± interquartile range

cilastatin's influence, if any, on AKI-CKD. We did not change the position of the animals every few hours as is performed in clinical care. As a result, even animals without impact treatment developed (later, less severe) crush injury due to prolonged immobility. We studied female pigs

only due to tractability; this limitation is offset to some extent by our prior work showing efficacy in male mice[18]. Nonetheless, these results may be limited to a single sex and/or strain of pigs. Lastly, although measurement of myoglobin clearance to confirm its role in the mechanism of cilastatin-

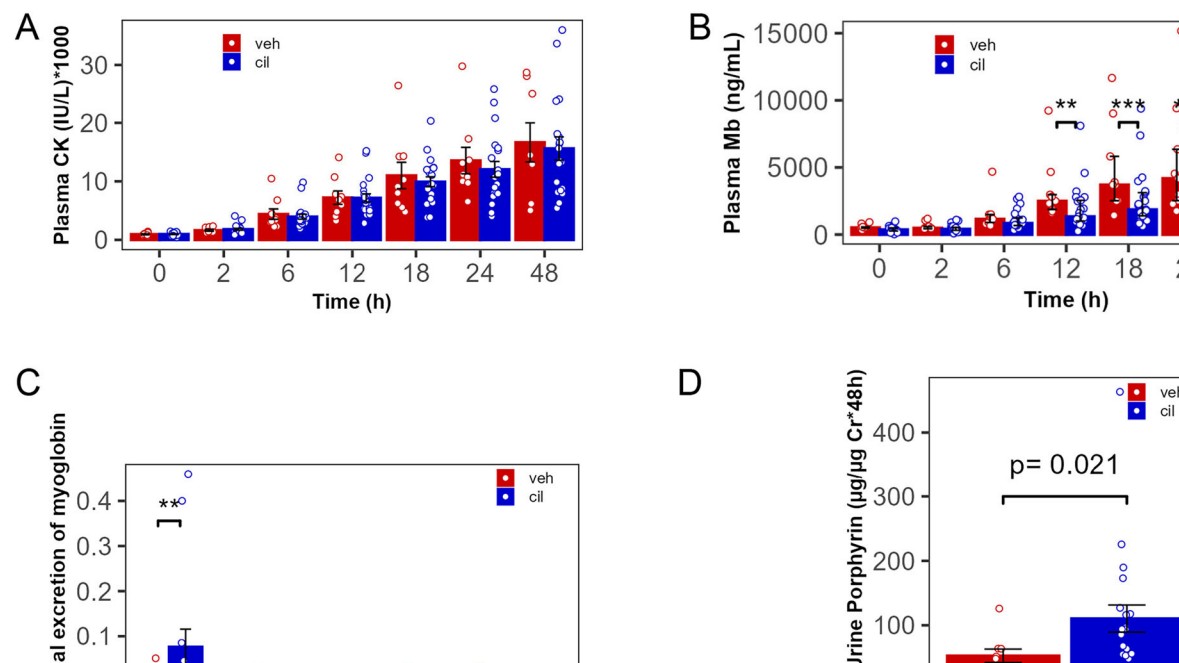

**Fig. 6 | Cilastatin increased urinary excretion of myoglobin.** Cilastatin sodium did not change plasma CK levels (**A**), suggesting it had little effect on muscle injury. However, Cil caused reduction in plasma myoglobin (Mb, $p = 0.0049$, <0.0001, <0.0001, respectively, at 12, 18, and 24 h, (**B**), suggesting alteration in the disposition of low molecular weight Mb, which undergoes renal filtration, different from high molecular weight CK, which does not undergo renal filtration. Fractional excretion of myoglobin was also increased ($p < 0.0001$ at 2 h) by cilastatin (**C**). To address the limited sensitivity of urine Mb measurement (see text), urine porphyrins were quantified; cilastatin greatly increased 48 h total porphyrin excretion (**D**) ($n = 9$ veh, 21 cil, *=$p < 0.05$, **=$p < 0.01$, **A**: linear mixed models regression, **B**, **C**: robust linear mixed models regression, **D**: Welch test. Error bars: **A**, **D** mean ± SEM, **B**, **C**: median ±interquartile range.

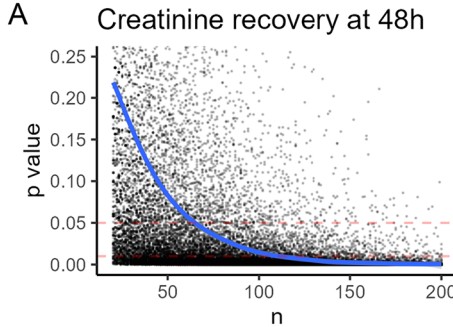
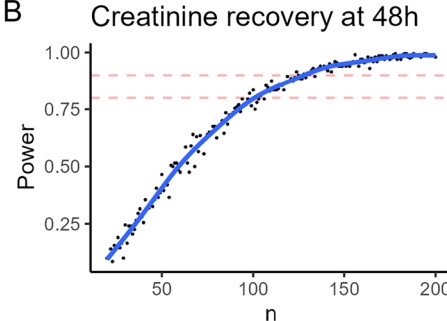

**Fig. 7 | Monte Carlo simulation-based power estimation for clinical trial of cilastatin sodium for crush syndrome using a patient-centered outcome.** Using input data from patients with crush syndrome and effect magnitude (recovery to <70% of maximum creatinine at 48 h) from these experiments, 500 clinical trials were simulated for each experimental number between 20 and 200. **A** $p$-value for each trial with the regression line. Each dot is one simulated trial, dashed red lines indicate 0.01 and 0.05 $p$-values. **B** Power calculation ([number of trials at each n with $p$ value < $\alpha$ value]/$n$). Each dot represents all the trials at that $n$. Dashed red lines indicate 80 and 90% power. 90% power at $\alpha = 0.01$ is provided by 124 patients (62 per group).

mediated renoprotection had reduced sensitivity, the corroborative data from porphyrin measurements are potentially limited by nonspecificity; these results should be interpreted in this light. Despite these limitations, this work provides strong evidence of the efficacy of cilastatin sodium to reduce kidney injury and the need for hyperkalemia treatment due to CS, in a highly translational model. Human studies will be required to determine whether these results are replicable in clinical settings, including austere conditions. Given the extensive safety record of cilastatin sodium and the outcomes of this large animal study, we believe there is every reason to hope that cilastatin may prove to have efficacy in future human studies.

## Data availability
Data required to reproduce the results reported here are shared on the Hutchens lab GitHub with digital object identifier 10.5281/zenodo.17478508[81]. The uniform resource locator (URL) for the repository is https://github.com/HutchensLab/Efficacy-of-cilastatin-sodium-in-large-animal[81].

## Code availability
Code required to reproduce the results, visualizations, and analysis reported here are shared on the Hutchens lab GitHub: https://github.com/HutchensLab/Efficacy-of-cilastatin-sodium-in-large-animal[81].

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

## Acknowledgements

The authors wish to acknowledge the concerted and dedicated assistance received from members of the OHSU clinical laboratory, especially Director Steven Kaczmierczak, and the OHSU analytical chemistry shared resource, especially Lisa Coussins. This work was supported by the US Department of Defense (W81XWH2010196/PR19304 to M.P.H) and the US Department of Veterans Affairs I01BX04288 (to M.P.H.) This material is the result of work which was supported with resources and the use of facilities at the Portland Veterans Affairs Medical Center. The contents do not represent the views of the U.S. Department of Veterans Affairs, the US Departement of Defense, or the United States Government.

## Author contributions

Conception, project direction, writing, editing: M.A.S. and M.P.H. Performance of animal experiments, assays, sample analysis, and project administration: J.P.G., K.M., M.B., S.D., LT-G, L.P., M.L., L.B., and L.L. Sample analysis, data analysis: M.B.E., J.F.H., and T.G. Critique, intellectual support, editing, writing: D.M. and I.J.S. Data analysis, data curation, codebase development, writing, editing: A.M. Pathologic analysis, writing, critique, figure preparation: N.K.A.

## Competing interests

Dr. Muruve is Co-founder and CSO of Arch Biopartners Inc, owner and licensee of the following patents on cilastatin for AKI. DPEP-1 binding compositions and methods of use. Robbins S, Senger D, Rahn J, Muruve DA, Lau A. US application 15/234,521 filed August 11, 2016, in reference to US provisional patent filed December 10, 2015. International application PCT/IB2016/001244 filed August 11, 2016. Use of cilastatin to reduce the nephrotoxicity of different compounds. Tejedor Jorge A, Lazaro Fernandez A, Camano Paez S, Torres Redondo AM, Lazaro Manero JA, Castilla Barba M, del Carmen De Lucas Collantes M. International application PCT/ES2008/070137 filed July 11, 2008. All other authors declare no competing interests.
