## [Transparent Peer Review file · Communications Medicine]

Efficacy of Cilastatin Sodium in a Translational Large Animal Crush Syndrome Model

Corresponding Author: Dr Michael Hutchens

Version 0:

Reviewer comments:

Reviewer #1

(Remarks to the Author)

Dear Editor

The article submitted by Michael Hutchens et al. entitled "Efficacy of Cilastatin Sodium in a Translational Large Animal Crush Syndrome Model", examine kidney damage induced by Crush Syndrome in large animals such as pigs. Is the second-most common cause of death in earthquakes, and a frequent cause of critical illness after burn, blast, and prolonged immobility. No specific treatment exists, and supportive treatment is highly burdensome, contributing to deaths in austere environments, especially disasters and conflicts. Therefore, there is urgent need for specific treatment which reduces burden of care. They investigate whether cilastatin sodium, has efficacy as a crush syndrome treatment in a highly translational large animal trauma model.

General Comments:

The manuscript presents solid and relevant data on the use of sodium cilastatin to mitigate crush syndrome in a porcine model. The methodology is robust, the animal model has high translational validity, and the results are clinically significant. Renal function measurement and physiological and biochemical evaluation are accurate. The discussion is well contextualized within the field of crush syndrome research. The work has the potential to lay the groundwork for human clinical trials and stands out for its focus on an underexplored clinical issue in disaster scenarios.

However, some aspects require clarification or further elaboration to strengthen the study's validity and applicability:

1. The authors state that "Crush syndrome is dependent on the renal megalin-dependent endocytic system," but this is a very general assertion. They should delve more into other contributing factors, such as the role of myoglobin and its degradation products in causing injury. The introduction barely discusses the pathophysiology of rhabdomyolysis underlying the syndrome. The authors attribute the injury solely to megalin-mediated myoglobin uptake, but the process is more complex. They fail to mention tubular obstruction or the direct toxicity of myoglobin breakdown products. They should expand this section to include more comprehensive information on the pathophysiology of Crush Syndrome.
2. They also state that cilastatin is a megalin inhibitor but provide little detail on its origin or its relationship with the enzyme DHP-I, as referenced in the article. Cited authors clearly indicate that the improvement in renal failure from any etiology is due to cilastatin's interaction with DHP-I, not solely to megalin inhibition. The authors should elaborate further and clarify the significance of these findings to avoid misinterpretation. As currently written, the injury and its improvement seem solely related to megalin.
3. Given cilastatin's central role in the article, it is insufficiently explored. Additional information should be included to emphasize its importance.
4. The authors indicate that cilastatin/vehicle was administered 30 minutes after injury induction, but they provide no explanation for this timing choice. A brief justification should be added.
5. They also state that differences between rodents and humans hinder translational applicability, providing several reasons in the text. While large animals indeed better reflect human disease than small animals, the current phrasing seems to downplay rodent data, even though the authors cite strong rodent-based studies supporting their conclusions. Although

cilastatin is not administered alone in humans, Zaballos et al. (2021) observed that imipenem/cilastatin improved cisplatin-induced AKI. The authors should reconsider the tone to avoid overly categorical statements.

6. The claim that rodent doses differ from approved human doses is overly definitive and incorrect. The doses used in the referenced cilastatin studies fall within the clinical range, especially when considering repeated doses and co-administration with imipenem. The doses used in animal models are pharmacologically relevant and translationally comparable to maximal clinical doses, particularly for sustained or protective effects in critical conditions. This statement should be rewritten to avoid confusion.

7. Regarding the dose used, the authors should explain the choice of 100 mg/kg, over three times the approved human daily dose. Although likely justified by interspecies differences, they should discuss whether this impacts translational relevance. The vehicle composition is also unspecified—could its components affect the study outcome? This is particularly important since significant changes were observed in the first hours post-administration. How do the authors assess potential vehicle effects?

Although they justify the use of a higher-than-FDA-approved dose and note no apparent acute toxicity, subacute or chronic effects were not explored, and follow-up lasted only 48 hours. This limits clinical generalizability, especially in patients with comorbidities. The discussion should include potential risks of dose escalation in humans and propose pharmacokinetic/pharmacodynamic studies. Using two cilastatin suppliers could introduce undetected differences, although enzymatic activity testing was performed

8. The authors should justify using only female pigs. While pigs are physiologically similar to humans, the exclusive use of females and a single crossbreed (Yorkshire-Landrace) may limit generalizability to other populations (e.g., males or other breeds).

9. Although the study is described as double-blinded, one could argue whether blinding was fully maintained, considering that uninjured animals may be easily distinguishable due to their more stable physiology. This point should be clarified. The 2:2:2:1 allocation ratio results in a smaller “uninjured” group, potentially reducing statistical power for certain comparisons.

10. The manuscript references supplemental figures, but they are not provided, making the text hard to follow. It is strongly recommended to include them for better understanding.

11. Cilastatin binds reversibly to its receptors, so the expected effect on vitamin D may be negligible or absent. The authors place too much emphasis on this analysis, which ultimately showed no relevance and appears to have been a non-significant pre-specified analysis. This could have been approached differently, perhaps focusing on other more relevant parameters. In the calcitriol group comparison, the authors should explain why a 1 µg dose was used and why it was administered in 125 mL. How does this affect study blinding if all animals receive this additional 125 mL?

12. Overall, the article contains too many sections and supplementary references, making it tedious to follow. The authors should make it more dynamic and improve readability.

13. The authors should explain the choice of the mean arterial pressure criterion. Several criteria are mentioned but not justified.

14. In the model, an IV fluid solution was administered immediately after injury. The authors should explain this choice and whether it may influence early results. Fluid resuscitation is not always immediate after crush injury, potentially affecting renal injury outcomes or sample parameters collected immediately post-injury.

15. Referencing the products used would help other researchers replicate the experiments.

16. Only one hematoxylin-eosin image is provided to demonstrate injury, and it is of poor quality. Additional histological images from all groups should be included, along with at least one KIM-1 staining to demonstrate tubular injury.

17. Regarding model limitations and follow-up duration: the conclusion of “no recovery” based on creatinine and GFR does not sufficiently establish chronic kidney disease risk. Renal recovery cannot be reliably inferred from 48-hour data. Early AKI reversal does not always predict long-term renal function. This limitation should be emphasized, and extended follow-up studies should be proposed. Extending the model to at least 72 hours without therapeutic interventions would help, as myoglobin-induced injury typically resolves by that time. Creatinine is already lower, and GFR higher at 48h, indicating a recovery trend.

18. Line 132 refers to calcitriol results in the introduction instead of in the results section, creating confusion. The inclusion of this group seems forced, especially since cilastatin acts transiently and its combination with calcitriol would not likely provide additional benefit if megalin is already blocked. The rationale should be better justified, or this section moved to an appendix.

19. While increased urinary myoglobin is proposed as a protective mechanism, the data are inconsistent due to matrix interference. Figure 6D significance is borderline. If the theory relies solely on this point, data should be repeated with more sensitive kits. Proteinuria values would be interesting. The use of porphyrins as an indirect marker is creative but should be discussed in terms of methodological limitations. Additional oxidative stress or cytotoxicity markers (e.g., 8-OHdG) could support the hypothesis. The authors should consider that cilastatin's effect may involve mechanisms beyond megalin

inhibition.

20. No control group received cilastatin alone. The study compares injured vs. non-injured groups but does not evaluate cilastatin's action in a control setting. The authors should justify this choice and ideally include at least a few animals for this comparison.

21. Rhabdomyolysis in non-injured animals is a source of confusion. Even uninjured animals reportedly developed mild rhabdomyolysis due to prolonged immobility, which reduces contrast between groups and may dilute treatment effects. This limitation should be discussed more explicitly. It could have been mitigated by repositioning animals or adding another control group.

22. The conclusion that impact causes crush syndrome is based mainly on increased potassium and CK; however, significant changes in creatinine and lactate are lacking, and GFR significance is borderline.

23. While the model is novel and clinically relevant, using a captive bolt gun with a cardboard pad may introduce variability in applied force. No mention is made of evaluating pain or suffering in animals, which may have ethical implications.

24. The "absence of calcitriol effect" section disrupts the already complex flow of the article. It should be moved to supplemental materials or reorganized. Results are non-significant and appear forced. The authors should clarify whether vitamin D levels were measured before and after calcitriol addition in all groups and whether urinary vitamin D-binding protein was considered. If not, they should consider including these values.

25. It would be interesting to perform a co-localization study of megalin with other proteins such as DHP-I.

26. Why was the enzymatic assay with cilastatin not performed in cell cultures? The procedure should be better explained.

27. GFR was not determined using serum/urinary creatinine at baseline, 24, and 48 hours to match iohexol-based estimates. Lacking actual baseline values is a limitation. The "normal" GFR was estimated from other studies, which may introduce bias due to individual differences. Although LC/MS/MS for iohexol is precise, its complexity could limit its use in early clinical studies.

28. Plasma calcitriol values show large variability not seen with cilastatin. In the graphs, the calcitriol group should have a distinct color as it is easily confused with "cil/veh" comparison. Groups should be renamed consistently with the main text.

29. The seasonal variation section is difficult to follow. Including animal sequential ID as a random factor is appropriate, but variability could have been minimized by conducting the experiment in a shorter or more uniform timeframe.

30. Terms like "no recovery" and "rapid reversal" are used interchangeably without clear definitions. Terminology should be standardized, and temporal/biochemical criteria clearly defined. Only one 2025 reference is cited; authors could provide a brief explanation of these criteria. A 70% drop in creatinine from peak at 48h is used as a recovery criterion, which may be arbitrary and not clinically validated. The clinical validity of this threshold should be discussed and possibly complemented with additional criteria (e.g., GFR recovery $\geq 75\%$).

31. Figures and Tables:

- Including a schematic figure of the proposed mechanism of action would help synthesize the study's conceptual model.

- Axis labeling across figures is inconsistent (e.g., UOP, MAP sometimes abbreviated, sometimes not). Axes should be standardized.

- Scale issues in several graphs make it hard to discern differences, especially where values vary widely. Authors should revise axes, especially in figures related to calcitriol. Figure 2 is referenced in Materials and Methods, but not appropriately addressed.

- Figure 3: the label "A)" is missing. The title appears at the top left of Figure 3 but is missing in other figures. This inconsistency should be corrected. Histological H&E images would help interpret Figure 3G.

- Figure 4: the authors consider 6h as an acute effect; they should explain this choice. Severe injury appears at 12h, and hyperkalemia is not significant until 24h. The conclusion of Figure 4 is unclear. Authors should clarify that the referenced data is from the impacted group in Figure 3. Cilastatin has a ~1h half-life and is excreted unmetabolized in urine. Although it may not be present in plasma after 12h, indirect or tubular accumulation effects may persist. Authors should explain how they ensure no cilastatin effect at later time points. The term "liquid intake" in Figure 4D seems inappropriate. The phrase "cil may have induced diuresis" is based on a single 60% increase in the first hour, not seen at other times. A more cautious phrasing is advised: "a transient increase in urine output was observed, potentially reflecting a diuretic effect."

- Authors should explain why those exclusion criteria were selected. Table legends are poorly explained. Clarify abbreviations

- Figure 5D only addresses necrosis, not cell infiltration, apoptosis, etc. Why were those scoring criteria used? Why weren't vascular or glomerular injury studied? The renal injury score could be more complete. Authors should explain why necrosis was emphasized (X2) and why other cell death types, such as apoptosis, weren't explored.

Reviewer's Conclusion:

This is a high-quality study with rigorous experimental design and relevant findings. With some adjustments in data interpretation, clearer articulation of the proposed mechanisms, and a more critical discussion of limitations, the manuscript

will be a valuable contribution to the literature on Crush Syndrome management.

Reviewer #2

(Remarks to the Author)

Crush syndrome develops several complications such as rhabdomyolysis, hyperkalemia, acidosis, acute kidney injury, disseminated intravascular coagulation, sepsis. No specific treatment has been established for patients with crush syndrome. This manuscript is well written. I read this manuscript with interest. There is a minor problem to revise. Therefore, please revise the original manuscript and answer the question below.

#1 In page9, line 182-184, the authors stated we conclude that cilastatin administration did not directly alter mean arterial pressure, heart rate, temperature, prothrombin time, plasma ionized calcium, or base excess. However, figure 4 legend shows that base excess was significantly increased at all time points starting 12h after cil administration.

Reviewer #3

(Remarks to the Author)

The authors investigated the effect of cilastatin sodium on AKI and hyperkalemia using a porcine crush syndrome model. This is a clinically relevant and important topic. However, several concerns should be addressed.

Major points:

1. The data distribution appears to be skewed in several analyses (e.g., Figures 5B and C and 6B and C). It seems that outliers may have influenced the mean values, making them appear high. The medians for creatinine, BUN, plasma MB, and porphyrin excretion do not appear to differ as much between the treatment and vehicle groups as they mentioned. I strongly recommend that the authors reconsider the data distribution and reanalyze accordingly.
2. What are the differences in serum creatinine and BUN levels between subjects with and without impact? The crush syndrome model employed may be too mild to detect a treatment effect of cilastatin sodium.
3. The authors used urinary porphyrin as a substitute for myoglobin due to technical constraints. Is porphyrin sufficiently specific to myoglobin to serve as a valid surrogate marker?
4. The study relies on repeated measurements using spot urine samples. In Figure 6C, I recommend presenting the fractional excretion of myoglobin (or porphyrin), calculated using plasma and urinary creatinine. Without creatinine correction, urinary porphyrin levels may be confounded by variations in urine dilution.
5. It is unclear how the authors analyzed 48-hour porphyrin excretion in Figure 6D despite using spot urine correction.

Minor points:

Supplemental Figures 1–5 are missing.

Please include a histological image of a non-impact subject.

Please provide the number of samples for each experimental group in the figure legends.

Version 1:

Reviewer comments:

Reviewer #1

(Remarks to the Author)

Dear Editor

The article submitted by Michael Hutchens et al. entitled "Efficacy of Cilastatin Sodium in a Translational Large Animal Crush Syndrome Model", examine kidney damage induced by Crush Syndrome in large animals such as pigs. Is the second-most common cause of death in earthquakes, and a frequent cause of critical illness after burn, blast, and prolonged immobility. No specific treatment exists, and supportive treatment is highly burdensome, contributing to deaths in austere environments, especially disasters and conflicts. Therefore, there is urgent need for specific treatment which reduces burden

of care. They investigate whether cilastatin sodium, has efficacy as a crush syndrome treatment in a highly translational large animal trauma model.

The manuscript presents solid and relevant data on the use of sodium cilastatin to mitigate crush syndrome in a porcine model. The methodology is robust, the animal model has high translational validity, and the results are clinically significant. Renal function measurement and physiological and biochemical evaluation are accurate. The discussion is well contextualized within the field of crush syndrome research. The work has the potential to lay the groundwork for human clinical trials and stands out for its focus on an underexplored clinical issue in disaster scenarios.

Reviewer's Conclusion:

After carefully evaluating the authors' responses and the revisions undertaken to address the requested points, I consider that the manuscript now meets the standards required for publication. The authors have provided clear and well-referenced explanations, and the necessary modifications have been appropriately implemented. I would like to commend the authors for the high quality of their work and note that the study holds substantial promise for translational relevance to clinical practice. I sincerely acknowledge and appreciate the significant effort invested by the authors.

Reviewer #2

(Remarks to the Author)

This revised version of the article is more refined according to my comments . So I have no further comments

Reviewer #3

(Remarks to the Author)

I greatly appreciate the authors' thoughtful consideration of my previous concerns and their thorough revision of the manuscript. I now have only a few minor comments:

1. Looking through the revised data (Figure 5A-C and 6C), I noticed that the effectiveness of cilastatin was limited. It did not consistently show a significant benefit across multiple timepoints. Therefore, I would recommend they avoid overpromising and tone down their claim.
2. Generally speaking, if data are distributed normally, mean \pm SEM may be fine (mean \pm SD is more appropriate). If data are distributed nonnormally, they should show median (interquartile range) in their bar graphs (or box plots).
3. Please show scale bars in their histological images or specify the magnification in the figure legends.

We are deeply grateful to our reviewers, whose comprehensive and thoughtful review has made our science better. Please see our point-by-point response below, with our responses preceded by ">>"

Reviewers' comments:

Reviewer #1 (Remarks to the Author):

Dear Editor

The article submitted by Michael Hutchens et al. entitled "Efficacy of Cilastatin Sodium in a Translational Large Animal Crush Syndrome Model", examine kidney damage induced by Crush Syndrome in large animals such as pigs. Is the second-most common cause of death in earthquakes, and a frequent cause of critical illness after burn, blast, and prolonged immobility. No specific treatment exists, and supportive treatment is highly burdensome, contributing to deaths in austere environments, especially disasters and conflicts. Therefore, there is urgent need for specific treatment which reduces burden of care. They investigate whether cilastatin sodium, has efficacy as a crush syndrome treatment in a highly translational large animal trauma model.

General Comments:

The manuscript presents solid and relevant data on the use of sodium cilastatin to mitigate crush syndrome in a porcine model. The methodology is robust, the animal model has high translational validity, and the results are clinically significant. Renal function measurement and physiological and biochemical evaluation are accurate. The discussion is well contextualized within the field of crush syndrome research. The work has the potential to lay the groundwork for human clinical trials and stands out for its focus on an underexplored clinical issue in disaster scenarios.

However, some aspects require clarification or further elaboration to strengthen the study's validity and applicability:

1. The authors state that "Crush syndrome is dependent on the renal megalin-dependent endocytic system," but this is a very general assertion. They should delve more into other contributing factors, such as the role of myoglobin and its degradation products in causing injury. The introduction barely discusses the pathophysiology of rhabdomyolysis underlying the syndrome. The authors attribute the injury solely to megalin-mediated myoglobin uptake, but the process is more complex. They fail to mention tubular obstruction or the direct toxicity of myoglobin breakdown products. They should expand this section to include more comprehensive information on the pathophysiology of Crush Syndrome.

>>We agree that the other causes of myoglobin-induced AKI were under-emphasized, however we are sharply limited in our ability to review the literature by the word count requirement. We have added a sentence which broadens the pathophysiology to the more commonly cited 3 etiologies of AKI, and referred the reader to several excellent review papers. We've been more specific in citing our own work which shows that GFR loss in heme-induced AKI is megalin-dependent, which was the rationale for the use of cilastatin.

2. They also state that cilastatin is a megalin inhibitor but provide little detail on its origin or its relationship with the enzyme DHP-I, as referenced in the article. Cited authors clearly indicate that the improvement in renal failure from any etiology is due to cilastatin's interaction with DHP-I, not solely to megalin inhibition. The authors should elaborate further and clarify the significance of these findings to avoid misinterpretation. As currently written, the injury and its improvement seem solely related to megalin.

>>We agree that cited authors say that improvement in renal failure from some etiologies is due to cilastatin's interaction with DHP-1, not solely due to megalin inhibition. We also agree that evidence supports DHP-1 interaction as a renoprotective factor in non-pigment nephropathies, and that DHP-1-dependent mechanisms may be part of the effect we observed in our swine and mouse studies. We reworded the section of the introduction to be more balanced, so that it now states: "Cilastatin sodium is US Food and Drug Administration-approved in combination with the antibiotic imipenem; it was identified as an inhibitor of renal dipeptidase (DHP-1) in the 1980's, and more recently as an inhibitor of megalin-mediated endocytosis (PMID 28052987). Cilastatin's renoprotective action in diverse rodent AKI models (PMIDs 39304269, 39025411, 35563891, 32899204, 26504822) is partly attributed to DHP-1 inhibition. It is also renoprotective in rhabdomyolysis-induced AKI by a megalin-dependent mechanism (PMID 34341182)."

We already discussed the limitations of our analysis of myoglobin excretion and the megalin-dependent mechanism in a paragraph of the discussion. We have added the following to that paragraph: "However it should be noted that cilastatin has demonstrated additional renoprotective properties (PMIDs 39304269, 39025411, 35563891, 32899204, 26504822, 33513824, 34830406, 28340076, 22718191, 20435919, 15252165) beyond increasing myoglobin clearance in crush syndrome, and it is possible these effects, believed to be exerted through interaction with renal dipeptidase, also contributed to the improved renal function that resulted from cilastatin administration."

3. Given cilastatin's central role in the article, it is insufficiently explored. Additional information should be included to emphasize its importance.

>>We gave additional context about cilastatin's development in the revised introduction. Given the necessity to be concise, we believe further review of cilastatin is beyond the scope of this paper. A review of cilastatin's role as a renoprotective agent is clearly indicated, and lacking in the literature. As a start, we respectfully suggest that if accepted, the present manuscript could be accompanied by an editorial giving additional background on the growing translational importance of cilastatin.

4. The authors indicate that cilastatin/vehicle was administered 30 minutes after injury induction, but they provide no explanation for this timing choice. A brief justification should be added.

>>We have added a brief justification to the manuscript. We are circumspect because additional data establishing this timepoint as an effective timepoint in mice (from our lab) are under review at another journal. For the reviewer's interest, in work documented in the manuscript under review, using the mouse glycerol model, we tested cilastatin administration timepoints from zero

to 4h after injury. Full efficacy was retained if the drug was administered 30 minutes after glycerol injection, and partial efficacy at 1h. Administration of cilastatin beyond 1h was not different from vehicle administration.

5. They also state that differences between rodents and humans hinder translational applicability, providing several reasons in the text. While large animals indeed better reflect human disease than small animals, the current phrasing seems to downplay rodent data, even though the authors cite strong rodent-based studies supporting their conclusions. Although cilastatin is not administered alone in humans, Zaballos et al. (2021) observed that imipenem/cilastatin improved cisplatin-induced AKI. The authors should reconsider the tone to avoid overly categorical statements.

>>We agree. We deleted the overly extensive commentary on mouse renal physiology and re-emphasized the importance of sus scrofa models in translational investigation.

6. The claim that rodent doses differ from approved human doses is overly definitive and incorrect. The doses used in the referenced cilastatin studies fall within the clinical range, especially when considering repeated doses and co-administration with imipenem. The doses used in animal models are pharmacologically relevant and translationally comparable to maximal clinical doses, particularly for sustained or protective effects in critical conditions. This statement should be rewritten to avoid confusion.

>>We deleted the statement.

7. Regarding the dose used, the authors should explain the choice of 100 mg/kg, over three times the approved human daily dose. Although likely justified by interspecies differences, they should discuss whether this impacts translational relevance. The vehicle composition is also unspecified—could its components affect the study outcome? This is particularly important since significant changes were observed in the first hours post-administration. How do the authors assess potential vehicle effects?.

>>The vehicle for cilastatin was PlasmaLyte A. We agree, the wording in the manuscript specifying this was overly terse and unclear. We have reworded the sentence. In the supplemental methods, the drug preparation method is extensively described. We have added the molar composition of PlasmaLyte A to the drug preparation section in the supplemental methods. Since the vehicle was present in all animals and was chosen because it is a standard resuscitation fluid, differential effects across groups should not be present.

Although they justify the use of a higher-than-FDA-approved dose and note no apparent acute toxicity, subacute or chronic effects were not explored, and follow-up lasted only 48 hours. This limits clinical generalizability, especially in patients with comorbidities. The discussion should include potential risks of dose escalation in humans and propose pharmacokinetic/pharmacodynamic studies.

>>We added this limitation to discussion: “Although a phase I trial (NCT03595189) including dosing similar to that used in our study was completed in 2018, results have not been published. Additional pharmacokinetic/pharmacodynamic study will therefore be required as part of translational research. “

Using two cilastatin suppliers could introduce undetected differences, although enzymatic activity testing was performed

>>We agree this is a limitation of the study as performed, and we performed activity testing to ensure equivalency of activity. Changes in preparation at any stage of the translational continuum could result in undetected differences.

8. The authors should justify using only female pigs. While pigs are physiologically similar to humans, the exclusive use of females and a single crossbreed (Yorkshire-Landrace) may limit generalizability to other populations (e.g., males or other breeds).

>> We added the following to the limitations paragraph of discussion: “We studied female pigs only due to tractability; this limitation is offset to some extent by our prior work showing efficacy in male mice (PMID 34341182), nonetheless, these results may be limited to a single sex and/or strain of pigs.”

9. Although the study is described as double-blinded, one could argue whether blinding was fully maintained, considering that uninjured animals may be easily distinguishable due to their more stable physiology. This point should be clarified. The 2:2:2:1 allocation ratio results in a smaller “uninjured” group, potentially reducing statistical power for certain comparisons.

>> We have clarified in the methods that surgeons were and the renal pathologist were blinded to drug treatment, and the pathologist was blinded to surgical model as well. We agree the 2:2:2:1 allocation reduced power for some comparisons. This is one reason we are cautious about conclusions regarding apparent differences between no-impact and impact-vehicle (see below).

10. The manuscript references supplemental figures, but they are not provided, making the text hard to follow. It is strongly recommended to include them for better understanding.

>>We apologize for the omission, these are provided in the resubmission.

11. Cilastatin binds reversibly to its receptors, so the expected effect on vitamin D may be negligible or absent. The authors place too much emphasis on this analysis, which ultimately showed no relevance and appears to have been a non-significant pre-specified analysis. This could have been approached differently, perhaps focusing on other more relevant parameters. In the calcitriol group comparison, the authors should explain why a 1 µg dose was used and why it was administered in 125 mL. How does this affect study blinding if all animals receive this additional 125 mL?

>> We appreciate the reviewer’s caution on the discussion of calcitriol. To reduce the overemphasis concern we removed the results section on “non effect of calcitriol” and simply refer to the supplementary analysis in a single sentence in the section on effects of cilastatin. For the reviewer’s interest, the concern about vitamin D stems from the considerable literature implicating vitamin D status in critical care outcomes, including AKI (e.g. PMIDs 34017850,

30294930, 28847615, 27369932, 26295008, 26356094, 25780095, 24557421, 24423347, 23945717). There is also a small amount of preclinical evidence that calcitriol may have benefit for rhabdomyolysis-induced AKI (PMID 31068635). During development of the project, reviewers at multiple levels raised the concern that acute vitamin D deficiency could worsen the primary outcome. Although randomized trials of acute vitamin D supplementation for AKI in critically ill patients have recently failed to prove benefit (e.g. PMID 40924491), these studies were not available at the time the present study was designed. Together these data argue in favor of inclusion of the calcitriol arm. The reviewer is correct that the result was nonsignificant. We chose the analysis provided as that was the planned analysis (pooling the calcitriol result if no difference was also planned) and we are certain of the result. We include the analysis of nondifference between calcitriol+cilastatin and cilastatin alone in supplementary analyses because it is necessary for rigorous documentation of the rationale for pooling the calcitriol+cilastatin and cilastatin groups. We are uncertain what more relevant parameters might be included but are amenable to adding limited additional analysis should the reviewer suggest it is critical. 1 µg is an effective clinically used dose for severe vitamin D deficiency. Given the weight difference between humans and pigs, we hypothesized this dose would be effective.

Since all animals received the 125 mL, surgeons remained blinded to the presence or absence of calcitriol, they were only aware that calcitriol treatment was a part of the study.

12. Overall, the article contains too many sections and supplementary references, making it tedious to follow. The authors should make it more dynamic and improve readability.

>>We thank the reviewer for this comment. We have reduced sections as described above and made other improvements to reduce complexity and increase readability.

13. The authors should explain the choice of the mean arterial pressure criterion. Several criteria are mentioned but not justified.

>>At the time of model development, despite a comprehensive literature search, we were unable to find an evidence-based mean arterial pressure (MAP) threshold for resuscitation in pig shock models. During model development we noted that animals with prolonged (≥ 20 minutes) unresuscitated $MAP < 40$ frequently developed irreversible shock and died, while resuscitation to higher MAP criteria resulted in frequent volume resuscitation. Therefore we chose the < 40 mmHg for ≥ 20 minutes criteria as consistent with severe shock, but survivable with resuscitation and minimizing bias from excessive crystalloid administration. We have added the following sentence to the "Management of hypotension" section of Supplementary Methods:

"These criteria were empirically determined during model development to avoid excessive mortality or excessive fluid administration."

14. In the model, an IV fluid solution was administered immediately after injury. The authors should explain this choice and whether it may influence early results. Fluid resuscitation is not always immediate after crush injury, potentially affecting renal injury outcomes or sample parameters collected immediately post-injury.

>>IV fluids were not administered until 30 minutes after injury. We found a sentence in the supplemental methods which ambiguously stated the time of initiation of resuscitation, and have corrected it to make clear that no intervention was provided until 30 minutes after injury.

15. Referencing the products used would help other researchers replicate the experiments.

>>We have added an additional table (**Supplementary Table 1**) with all reagents and products mentioned in the manuscript.

16. Only one hematoxylin-eosin image is provided to demonstrate injury, and it is of poor quality. Additional histological images from all groups should be included, along with at least one KIM-1 staining to demonstrate tubular injury.

>>We have added an additional supplemental figure (**Supplementary Fig. 4**) with images from all groups. Although we tried several antibodies and a wide variety of staining protocols, we have been unable to successfully stain KIM-1 in the porcine kidney. Review of the literature reveals a paucity of publications in which investigators demonstrate KIM-1 staining in the porcine kidney. Given the specificity of the findings on PAS (including casts) and measurement of GFR, we respectfully do not believe that staining for biomarkers of kidney injury adds significant information to the manuscript.

17. Regarding model limitations and follow-up duration: the conclusion of “no recovery” based on creatinine and GFR does not sufficiently establish chronic kidney disease risk. Renal recovery cannot be reliably inferred from 48-hour data. Early AKI reversal does not always predict long-term renal function. This limitation should be emphasized, and extended follow-up studies should be proposed. Extending the model to at least 72 hours without therapeutic interventions would help, as myoglobin-induced injury typically resolves by that time. Creatinine is already lower, and GFR higher at 48h, indicating a recovery trend.

>>Please see our response to item #30 below for more detail about “no recovery” outcomes. We have changed the relevant section of discussion/limitations to read:

“Additionally, although AKI was resolving at 48h and cilastatin increased the number of animals with improvement in renal function at that time point, longer-term outcomes of crush syndrome or effects of cilastatin beyond 48h were not assessed and reduced risk of development of CKD cannot be inferred. Longer-term studies should be performed to evaluate cilastatin’s influence, if any, on AKI-CKD.”

We also changed the wording of results and methods sections relevant to this measurement to be much more circumspect about long-term inference. The relevant methods section now reads:

“Increasing preclinical and clinical data suggest that preclinically-defined renal repair, maladaptive repair, and AKI-CKD transition are likely associated with clinical AKI recovery trajectories including rapid recovery, delayed recovery, and nonrecovery (Reviewed in Ostermann, Forni, Joannidis, Kane-Gill, Legrand, Lumlertgul, McNicholas, Meersch, Monard, Pickkers, Prowle, Rimmelé, Schneider, Zarbock and Kellum 40719889). These trajectories are also clinically associated with mortality and remote organ dysfunction. To assess this important translational outcome, we identified relevant thresholds from the clinical literature for rapid

recovery at 48h and applied them. Consensus guidelines define persistent AKI as that which continues beyond 48h. Applicable (within 48h) clinical thresholds for rapid recovery which reduced risk of dialysis, mortality, or development of chronic kidney disease were identified as 30% recovery of creatinine (PMID 38765592) and 2) failure to recover to 70% of baseline eGFR (PMID 23124779). To enable powering for a future clinical trial from this 48h study, we performed Monte Carlo simulation using the categorical variable of recovery to 70% of the maximal creatinine by 48h as the outcome. “

18. Line 132 refers to calcitriol results in the introduction instead of in the results section, creating confusion. The inclusion of this group seems forced, especially since cilastatin acts transiently and its combination with calcitriol would not likely provide additional benefit if megalin is already blocked. The rationale should be better justified, or this section moved to an appendix.

>> We removed this sentence from the introduction and placed it in the supplementary analysis. We thank the reviewer for making this text clearer.

19. While increased urinary myoglobin is proposed as a protective mechanism, the data are inconsistent due to matrix interference. Figure 6D significance is borderline. If the theory relies solely on this point, data should be repeated with more sensitive kits. Proteinuria values would be interesting. The use of porphyrins as an indirect marker is creative but should be discussed in terms of methodological limitations. Additional oxidative stress or cytotoxicity markers (e.g., 8-OHdG) could support the hypothesis. The authors should consider that cilastatin's effect may involve mechanisms beyond megalin inhibition.

>>We agree with the reviewer. As discussed above in this response and in the revised discussion of the manuscript we do not provide conclusive evidence that the myoglobin excretion mechanism is the mechanism of protection in this manuscript. In rodents, which have constitutive proteinuria and in which it is feasible to collect and cryostore complete 24h urine samples, we demonstrated this mechanism was 10x greater in cilastatin-treated animals, and that renoprotection was megalin-dependent (that is, the protective effect of cilastatin is lost in the absence of megalin, figure 6 of PMID 34341182). As above, we agree with the reviewer that other protective mechanisms may be in place, and this has been documented in the discussion. The specific hypothesis tested in this manuscript is that cilastatin is effective to ameliorate AKI in a large animal model. We evaluated mechanism only to confirm that myoglobin clearance is increased in this model, consistent with results in mice. In the end, likely due to matrix interference, we were unable to reliably measure urine myoglobin in pigs. Therefore we quantified total porphyrins; this demonstrated that cilastatin increased urine porphyrins. There are 3 significant potential sources of urine porphyrins: bilirubin, hemoglobin, and myoglobin. Because hemoglobin and myoglobin are both released by crush injury (crush damages blood vessels which remain intact, resulting in ongoing hemolysis) hemolysis is a component of crush syndrome and not a confounder. Bilirubin is a potential confounder, if cilastatin caused or prevented liver injury. We have no indication of liver injury in this model (specifically, prothrombin time was not altered, figure 4F), however, we agree that nonspecificity of total porphyrins is a limitation and we have included a statement to that effect in the discussion section:

“Lastly, although measurement of myoglobin clearance to confirm its role in the mechanism of cilastatin-mediated renoprotection had reduced sensitivity, the corroborative data from porphyrin measurements is potentially limited by nonspecificity; these results should be interpreted in this light.”

20. No control group received cilastatin alone. The study compares injured vs. non-injured groups but does not evaluate cilastatin’s action in a control setting. The authors should justify this choice and ideally include at least a few animals for this comparison.

>>The reviewer is correct that we did not include a cilastatin+no injury group. The hypothesis in this study was that cilastatin would ameliorate AKI caused by crush injury. The specific test of that hypothesis is comparison between cilastatin and vehicle in crush-injured animals. Therefore a cilastatin+no surgery group would not test the hypothesis. Nonetheless, at the time the study was designed, we planned a cilastatin+no surgery study to assess cilastatin effects in a control setting. Shortly after our surgical study was started however, a phase I dose-escalation study in humans, which included doses similar to the one we delivered to pigs, completed (NCT03595189). The combination of the existence of human data and that our planned cilastatin+no surgery study did not directly test our hypothesis meant that it was no longer in accordance with the animal research ethical principle of reduction for us to perform the cilastatin+no surgery study.

21. Rhabdomyolysis in non-injured animals is a source of confusion. Even uninjured animals reportedly developed mild rhabdomyolysis due to prolonged immobility, which reduces contrast between groups and may dilute treatment effects. This limitation should be discussed more explicitly. It could have been mitigated by repositioning animals or adding another control group.

>>We agree with the reviewer that this result could be confusing. We were unable to safely change the position of anesthetized pigs because of their instrumentation. We are unsure whether positioning change would have been effective. Although this length of time without repositioning is unusual in human intensive care, it is not unheard of, it likely occurs in critical care in austere circumstances, where lifesaving tasks are prioritized and nursing care may be reduced, and it occurs during prolonged intraoperative care. The comparison we make is between two groups with equal probability of the rhabdomyolysis outcome, with the exception of drug treatment, and therefore the result we report is a valid one. We specifically report this limitation in the discussion paragraph delineating limitations as follows:

“We did not change the position of the animals every few hours as is performed in clinical care. As a result, even animals without impact treatment developed (later, less severe) crush injury due to prolonged immobility.”

22. The conclusion that impact causes crush syndrome is based mainly on increased potassium and CK; however, significant changes in creatinine and lactate are lacking, and GFR significance is borderline.

>>We agree with the reviewer’s statement. We respectfully note that consensus research guidelines defining crush syndrome are lacking. Bywaters’ seminal description included shock, hyperkalemia, cardiac arrest, and pigment nephropathy. Clinical reviews typically define crush syndrome as crush injury with systemic manifestations which include kidney injury or failure,

hyperkalemia, hypocalcemia, and acidosis (eg PMID 40459168,14979336). We used 2 definitions of crush syndrome in designing and evaluating outcomes of the study.

The International Society of Nephrology, Renal Disaster Relief Task Force “Recommendations for the Management of Crush Victims in Mass Disasters” (PMID 22467763), referencing Bywaters and one of the reviews cited above, defines crush syndrome as “Crush injury and systemic manifestations due to muscle damage [16, 21, 22]. Systemic manifestations may include acute kidney injury (AKI), sepsis, acute respiratory distress syndrome (ARDS), disseminated intravascular coagulation (DIC), bleeding, hypovolemic shock, cardiac failure, arrhythmias, electrolyte disturbances and psychological trauma [20, 23].”

US Department of Defense Joint Trauma System Clinical Practice Guideline, “Management of Crush Syndrome under Prolonged Field Care” (PMID 27734449) defines crush syndrome as “... a reperfusion injury that leads to traumatic rhabdomyolysis. Reperfusion results in the release of muscle cell components, including myoglobin and potassium, that can be lethal. Myoglobin release results in rhabdomyolysis, with risk of kidney damage. Kidney damage leads to hyperkalemia and eventually cardiac arrhythmias. Calcium is taken up by injured muscle cells and this can cause hypocalcemia, contributing to cardiac arrhythmias. “

Therefore we tested the hypothesis that injured animals developed crush syndrome as defined by elevated CK, hyperkalemia, hypocalcemia, and kidney injury. The outcomes we report, including histopathology-based evaluation of kidney injury support this definition and our conclusion.

23. While the model is novel and clinically relevant, using a captive bolt gun with a cardboard pad may introduce variability in applied force. No mention is made of evaluating pain or suffering in animals, which may have ethical implications.

>>We agree with the reviewer that variability of force applied is a challenge. We addressed this challenge by quantifying CK and assessing change in CK from baseline to after injury – animals with minimal change in CK were excluded. This is documented in Figure 1, Supplementary Table 2 and in the Results Section as follows:

“In 6 animals (3 vehicle-, 3 cilastatin-treated), despite apparent impact, CK was not elevated by 6h (baseline to 6h change in CK: 27 ± 10 IU/kg vs. 61 ± 4 IU/kg, $p=0.0012$). Based on these *a priori* exclusion criteria, these animals were excluded from further analysis (**Supplementary Table 2**).”

In response to the reviewer’s concern, on review of the Supplemental Methods section on anesthesia, we noted 2 omissions. We omitted that animals received single-dose buprenorphine once the intravenous catheter was placed, and second, the dose for ketamine was given as fixed, when actually it was titrated to absence of specific signs of discomfort. We corrected the omissions and, with deference to the importance of the reviewers impression, edited this section to ensure it is clear we considered discomfort. The section now reads:

“To provide analgesia, buprenorphine 0.01 mg/kg was administered via the jugular catheter... The animal received ketamine via continuous infusion for augmentation of maintenance

anesthesia and analgesia at a dose of 1-5 mg/kg/hr; this was titrated to ensure the absence of signs of discomfort.”

We respectfully disagree with the statement that no mention was made of evaluating pain or suffering in animals. Efforts to address animal pain and suffering were extensive. Ethical compliance was documented in the Methods section (“All procedures were approved by the Institutional Animal Care and Use Committee of Oregon Health & Science University, and the Animal Care and Use Review Office of US Army Medical Research and Development Command.”). Animals were under general anesthesia for the entire duration of procedures.

24. The “absence of calcitriol effect” section disrupts the already complex flow of the article. It should be moved to supplemental materials or reorganized. Results are non-significant and appear forced. The authors should clarify whether vitamin D levels were measured before and after calcitriol addition in all groups and whether urinary vitamin D-binding protein was considered. If not, they should consider including these values.

>> We removed reference to the absent effect of calcitriol (as discussed above) from the results except for a brief reference to the supplementary explanation. Vitamin D levels were measured before and after calcitriol administration, but the baseline values were previously omitted from the supplemental file. We revised the supplemental file to include them. We did not measure calcitriol levels in no-impact animals, as that measure was not in the pre-planned analysis to test the impact of calcitriol administration on the effect of cilastatin. The results provided are the data we obtained and are provided in a format meant to provide comprehensive and rigorous documentation of the analysis we performed. We agree the results are nonsignificant, and the finding of nondifference is important to the overall study, but belong, as the reviewer suggests, in the supplementary materials. We did not measure vitamin D binding protein in the urine.

25. It would be interesting to perform a co-localization study of megalin with other proteins such as DHP-I.

>>We appreciate the reviewer’s interest in this important question. Megalin and DHP-1 (renal dihydropeptidase, also known as renal dipeptidase, DPEP-1, e.g. Uniprot accession #P16444) are both robustly expressed in the renal brush border and have been extensively imaged in our work (for example PMID 34341182) and that of others (for example Lau et al, Science Advances 2022, PMID 35108057). Because this area has rich and overlapping transporter expression, it is very likely that DPEP1 and megalin colocalize on the brush border, and high-resolution microscopy, likely super-resolution microscopy would be necessary to spatially resolve their separation. We agree that given their spatial and perhaps functional overlap such studies are necessary; however given the limited availability of well-characterized antibodies in sus scrofa, mouse or human studies would be more likely to succeed. Regrettably, such studies are beyond the scope of this manuscript.

26. Why was the enzymatic assay with cilastatin not performed in cell cultures? The procedure should be better explained.

>>We chose a well-characterized assay which uses reagents of known and relatively invariable activity and concentration to determine the inhibitory activity of preparations of cilastatin at renal

dipeptidase. The preparations were compared in terms of certificate of analysis data (i.e., purity by HPLC), as well as vendor-provided solubility and other data as available. This left open the important question of chemical activity at the target model. Since the experimental question was whether preparations from different suppliers differed in this key chemical property, we believe it was appropriate to use an *in chemico* assay. This assay has been widely-used and published by the performing author's lab (DM, PMIDs 35108057, 31442408). Use of cell culture would add variability; we believe this variability would be appropriate if there were questions about trafficking of the drug or of dipeptidase itself. We did not think these questions were within the scope of the present manuscript.

27. GFR was not determined using serum/urinary creatinine at baseline, 24, and 48 hours to match iohexol-based estimates. Lacking actual baseline values is a limitation. The "normal" GFR was estimated from other studies, which may introduce bias due to individual differences. Although LC/MS/MS for iohexol is precise, its complexity could limit its use in early clinical studies.

>>Our study was designed to measure GFR using iohexol because this is the most rigorous and widely validated method available for large animals at the time the study was designed. We also determined GFR using serum creatinine, urine creatinine, and urine output (instantaneous creatinine clearance) at all time points for which this data was available. As the reviewer undoubtedly knows this measurement can be confounded, especially during AKI when GFR can be changing. We do not report creatinine-based GFR because it is not a good estimation of measured GFR in this model. To determine whether we could use creatinine-based GFR, we performed Bland-Altman analysis on all matched measurements (that is, at 6, 24, and 48h). At 6 h, the Bland-Altman bias was 17 (creatinine clearance overestimates GFR by 17 mL/min), however the limits of agreement were very broad (-67 to 102 ml/min) meaning that 95% of the differences in measurement fell within this range. Given the range of the measurement overall, this is very poor agreement, and creatinine clearance would not support careful inference about GFR. There was heteroscedasticity as well: as the mean of both methods increases, agreement worsens; which is to say, the methods agree best when GFR is very low. We confirmed this was so with analysis at specific time points. We think, since we did not present the creatinine clearance in the manuscript, this supplementary analysis is really beyond the scope, and we did not include it in the supplement. However we are happy to do so if the reviewer wishes. We agree that lacking baseline GFR is a limitation, and that iohexol GFR is too cumbersome for clinical use. We are encouraged by recent FDA approval of a drug/device combination which allows fluorescence-based GFR measurement in humans, and we are currently designing studies using this device.

28. Plasma calcitriol values show large variability not seen with cilastatin. In the graphs, the calcitriol group should have a distinct color as it is easily confused with "cil/veh" comparison. Groups should be renamed consistently with the main text.

>>We thank the reviewer for pointing out this discontinuity in color design in the supplemental analysis. We reformatted the relevant supplemental analysis as discussed above, and now provide a consistent color scheme with cilastatin alone and cilastatin with calcitriol denoted in distinct colors from the red and blue used elsewhere in the manuscript to denote different groupings.

29. The seasonal variation section is difficult to follow. Including animal sequential ID as a random factor is appropriate, but variability could have been minimized by conducting the experiment in a shorter or more uniform timeframe.

>>We agree that higher experimental throughput might have reduced some seasonal variation; we suspect it might have introduced other variation however. For example, the cadence of experimental work was deliberate and allowed for careful analysis of surgical events and data, ensuring protocol adherence. Consistency of animal housing and surgical suite use was also facilitated by the chosen pace of procedures; had we done more animals/time we would have needed additional staff, a second operating room and additional preoperative and postoperative housing – all of which could have introduced variation. Although we do not include the analysis in the manuscript, we did perform analysis to determine whether the operative surgeon or assistant associated with outcomes – they did not, and we take this as evidence that protocol and surgical training were consistent.

We apologize for the difficulty in following the supplementary analysis of seasonal variation. We have added explanation and rewritten somewhat to improve readability. We include this supplementary analysis because we believe it is necessary to demonstrate rigor, and because the choice of how to account for the variation was complex and carefully considered. We only include code or code results as required for explication.

30. Terms like “no recovery” and “rapid reversal” are used interchangeably without clear definitions. Terminology should be standardized, and temporal/biochemical criteria clearly defined. Only one 2025 reference is cited; authors could provide a brief explanation of these criteria. A 70% drop in creatinine from peak at 48h is used as a recovery criterion, which may be arbitrary and not clinically validated. The clinical validity of this threshold should be discussed and possibly complemented with additional criteria (e.g., GFR recovery $\geq 75\%$).

>>Acute kidney disease (AKD), recovery from AKI, and the relationship between recovery and early-onset CKD are together a rapidly-expanding area of investigation. Because of the newness of the field, there is not consensus on what constitutes “rapid reversal” or “early recovery”, with one recent review providing a table of more than 20 different clinical outcome definitions (PMID 40719889). Basic/translational reversal/recovery definitions are equally challenged. A consensus group (the XXXVII Acute Disease Quality Initiative on Targeting Persistent Acute Kidney Injury and Promoting Recovery, <https://www.adqi.org/>) has been empanelled and will convene in Spring 2026, and it reasonable to expect published definitions within the following year.

We agree we did not clarify terms well in the methods section. We have rewritten both Supplemental and manuscript methods sections regarding this outcome.

31. Figures and Tables:

- Including a schematic figure of the proposed mechanism of action would help synthesize the study’s conceptual model.

>>We now provide an additional figure explaining the putative mechanism and potential other mechanisms (supplementary figure 6D)

- Axis labeling across figures is inconsistent (e.g., UOP, MAP sometimes abbreviated, sometimes not). Axes should be standardized.

>>We have standardized axis labeling, thank you for spotting the inconsistency.

- Scale issues in several graphs make it hard to discern differences, especially where values vary widely. Authors should revise axes, especially in figures related to calcitriol. Figure 2 is referenced in Materials and Methods, but not appropriately addressed.

>>We revised the referenced axes, including the calcitriol figure in supplementary analysis. We expanded the reference to figure 2, (which is a diagram of the model) in the methods section

- Figure 3: the label "A)" is missing. The title appears at the top left of Figure 3 but is missing in other figures. This inconsistency should be corrected. Histological H&E images would help interpret Figure 3G.

>>We fixed the label and title issues. Histological images have been added.

- Figure 4: the authors consider 6h as an acute effect; they should explain this choice. Severe injury appears at 12h, and hyperkalemia is not significant until 24h. The conclusion of Figure 4 is unclear. Authors should clarify that the referenced data is from the impacted group in Figure 3. Cilastatin has a ~1h half-life and is excreted unmetabolized in urine. Although it may not be present in plasma after 12h, indirect or tubular accumulation effects may persist. Authors should explain how they ensure no cilastatin effect at later time points. The term "liquid intake" in Figure 4D seems inappropriate. The phrase "cil may have induced diuresis" is based on a single 60% increase in the first hour, not seen at other times. A more cautious phrasing is advised: "a transient increase in urine output was observed, potentially reflecting a diuretic effect."

>>We regret the error in the caption to figure 4; we assessed cardiovascular and non-renal outcomes for this measure ("acute/direct drug effect") up to 12h. We have corrected this to match the results, which have been revised to state the reasoning. This time point was chosen as >2x the maximal expected plasma dwell time for cilastatin (5 half-lives ~ 6 hours), accounting for up to halving of renal function. The goal of this measure was to perform secondary analysis for signals of toxicity, which we find lacking. The reviewer is correct that secondary or delayed effects were not accounted for in this analysis. We do not ensure that off-target effects are absent at later time points; this is beyond the scope of this translational efficacy trial. The endpoints chosen here were chosen for the purpose of informing phase II clinical trials. For example, the potential for diuresis, which has not been reported before, may indicate unknown kidney effects, and is important to have available in design of human studies. We have revised the statement in the text to match the reviewer's preferred wording.

- Authors should explain why those exclusion criteria were selected. Table legends are poorly explained. Clarify abbreviations

We have rewritten the table's explanatory text to explain the choice of exclusion criteria and provide better explanation of the table, which has also been reformatted.

- Figure 5D only addresses necrosis, not cell infiltration, apoptosis, etc. Why were those scoring criteria used? Why weren't vascular or glomerular injury studied? The renal injury score could be more complete. Authors should explain why necrosis was emphasized (X2) and why other cell death types, such as apoptosis, weren't explored.

>>The goal of histopathologic outcomes in this study was quantification of any differential injury between groups, in order to test the stated hypothesis. Because pigment nephropathy preferentially injures the tubular epithelium (glomerular and vascular pathologic damage is not usually seen in this setting) we used a published tubular damage score. For brevity, we cited a recent, high-impact use of the score to evaluate histopathologic injury in rhabdomyolysis in our manuscript methods section (PMID 39356748) but omitted that citation from the Supplementary Methods. We have placed the citation in the Supplementary methods. The double-weighting of necrosis improves sensitivity to the most abundant and pathognomonic cell death pathway of AKI. We note apoptosis and regulated cell death pathways are not readily quantifiable on H&E stained slides. We did not explore other mechanisms of cell death because that mechanistic study is beyond the scope of this investigation and likely better conducted in an animal with more widely available immunoassays and antibodies.

Reviewer's Conclusion:

This is a high-quality study with rigorous experimental design and relevant findings. With some adjustments in data interpretation, clearer articulation of the proposed mechanisms, and a more critical discussion of limitations, the manuscript will be a valuable contribution to the literature on Crush Syndrome management.

Reviewer #2 (Remarks to the Author):

Crush syndrome develops several complications such as rhabdomyolysis, hyperkalemia, acidosis, acute kidney injury, disseminated intravascular coagulation, sepsis. No specific treatment has been established for patients with crush syndrome. This manuscript is well written. I read this manuscript with interest. There is a minor problem to revise. Therefore, please revise the original manuscript and answer the question below.

#1 In page9, line 182-184, the authors stated we conclude that cilastatin administration did not directly alter mean arterial pressure, heart rate, temperature, prothrombin time, plasma ionized calcium, or base excess. However, figure 4 legend shows that base excess was significantly increased at all time points starting 12h after cil administration.

>>We apologize for the lack of clarity. This figure refers to direct/acute effects of the drug, which we evaluated during the first 12 hours, because that is more than twice the expected time for complete clearance of cilastatin even in animals with severe kidney impairment. This time limit was chosen to represent a conservative estimate of the maximal plasma residence time for cilastatin sodium, as the plasma half-time is estimated as 45-60 minutes (FDA briefing document from 1989, and PMID 2571484) and 12h accounts for >5 half times even in the

presence of >50% GFR loss. We revised the caption for figure 4 to include this information and also updated the results section to specify the reason for the 12h time point.

Reviewer #3 (Remarks to the Author):

The authors investigated the effect of cilastatin sodium on AKI and hyperkalemia using a porcine crush syndrome model. This is a clinically relevant and important topic. However, several concerns should be addressed.

Major points:

1. The data distribution appears to be skewed in several analyses (e.g., Figures 5B and C and 6B and C). It seems that outliers may have influenced the mean values, making them appear high. The medians for creatinine, BUN, plasma MB, and porphyrin excretion do not appear to differ as much between the treatment and vehicle groups as they mentioned. I strongly recommend that the authors reconsider the data distribution and reanalyze accordingly.

>>We are deeply grateful for this critique, which caused us some concern, and which has resulted in a better manuscript. After careful review of all results, we agree with the reviewer that mean values in the referenced figures and some others were likely skewed by outlier values. While linear mixed models have been reported to have some robustness to violation of distributional assumptions, this robustness has limits (DOI 10.1111/2041-210X.13434). We had already undertaken efforts to reduce variation and outliers/skewness by careful animal model design and with exclusions for abnormal baseline status (Figure 1 in the manuscript, and supplemental table of exclusions). Therefore we reviewed the literature and consulted with a statistician before proceeding. After consultation we chose to approach distributional skewness/kurtosis with a surveillance approach. We reviewed all presented results for the presence of outliers or apparently non-normal distribution. For each figure in which it appeared possible that outliers skewed the distribution of data we examined residuals using histograms, residual vs. predicted value plots, and quantile (or “q-q”) plots. When these instruments supported a strong influence from outliers or non-normal distribution, we performed reanalysis of the data using outlier-robust linear mixed models (R package *robustlmm* DOI: 10.18637/jss.v075.i06).

Reanalysis was necessary for the following figures, with the described results. Figures 3A and 3C – additional time points are significantly different. Figures 6B (plasma myoglobin) and Figure 6C (previously urine/plasma myoglobin, now presented as fractional excretion of myoglobin as requested by this reviewer) – in 6B, the number of time points at which statistical significance was reached increased from 2 to 3. In 6C, the number of significant time points decreased from 2 to 1. Figures 5B and 5C (creatinine, number of significant time points was reduced to 1 from 2, and BUN, previously not shown). Urine total protein (previously not presented, now presented in supplemental in response to a request from reviewer 1) required robust analysis. We have revised the text and methods to reflect the statistical approach and the change in results. Because the affected results are corroborative to the primary outcome, and results remain statistically significant, our conclusions resulting from the data are not changed.

2. What are the differences in serum creatinine and BUN levels between subjects with and

without impact? The crush syndrome model employed may be too mild to detect a treatment effect of cilastatin sodium.

>>With respect for the reviewer's concern, the model is not too mild; during development we performed impact-titration, and at more than 4 total thigh impacts the mortality was too high to complete the study. We provide the no-impact model not to show a non-treated healthy baseline, but to control for the result of impact. We respectfully point out that histologic damage was more than 3 times higher in impacted animals (vehicle) than in no-impact animals (Figure 3G), demonstrating that there is an important, kidney-relevant detectable difference between no-impact and vehicle. BUN and creatinine are presented in supplementary figure 3. They are not different between sham and vehicle although they trend higher in impacted animals. We note the reviewer's concern that the no-impact and impact models are not very different, but this is not due to their mild nature. It is due to the stasis-related development of rhabdomyolysis in no-impact animals, which occurred to a lesser extent and later in no-impact animals compared with impacted animals. While it is potentially confounding that no-impact animals developed crush syndrome (discussed as a limitation in the manuscript) it does not detract from their value as a descriptive control for the model. The main comparison in this manuscript is between cilastatin and vehicle-treated impacted animals.

3. The authors used urinary porphyrin as a substitute for myoglobin due to technical constraints. Is porphyrin sufficiently specific to myoglobin to serve as a valid surrogate marker?

>>We appreciate the question. Urine myoglobin measurements were insensitive. Therefore we used urine porphyrin detection as a corroborative measurement. Because the limitation of urine myoglobin was insensitivity, a more sensitive, and therefore less specific method was appropriately chosen. Urinary porphyrins may reflect bilirubin and other enteric pigments, hemoglobin, or myoglobin, and could be confounded by group differential changes in these parameters. Because urine porphyrin measurements are confirmative of myoglobin measurements however, we do not think that confounding of porphyrin is a likely limitation.

4. The study relies on repeated measurements using spot urine samples. In Figure 6C, I recommend presenting the fractional excretion of myoglobin (or porphyrin), calculated using plasma and urinary creatinine. Without creatinine correction, urinary porphyrin levels may be confounded by variations in urine dilution.

>> We now provide fractional excretion of myoglobin as figure 6C as requested by the reviewer (calculated as $100 \times (SCr \times UMb) / (SMb \times UCr)$). Figure 6D has been remade to take urine creatinine into account. Supplemental methods has been updated to reflect these changes.

5. It is unclear how the authors analyzed 48-hour porphyrin excretion in Figure 6D despite using spot urine correction.

>>The AUC was constructed by numerical integration using Simpson's rule (R package Bolstad 2). This is a standard method for AUC estimation using time-separated y values. Supplemental methods has been updated to reflect this method.

Minor points:

Supplemental Figures 1–5 are missing.

>>We regret the error, supplemental figures are all available in the revision.

Please include a histological image of a non-impact subject.

>>Figure 2G now includes an exemplary image of a non-impact subject.

Please provide the number of samples for each experimental group in the figure legends.

>>We have added this information to the figure legends.

Author's response to reviewers:

We are again grateful to our reviewers for the thoughtful and detailed critique. Each criticism is followed by individual response immediately below, preceded by ">>".

Reviewers' comments:

Reviewer #1 (Remarks to the Author):

Dear Editor

The article submitted by Michael Hutchens et al. entitled "Efficacy of Cilastatin Sodium in a Translational Large Animal Crush Syndrome Model", examine kidney damage induced by Crush Syndrome in large animals such as pigs. Is the second-most common cause of death in earthquakes, and a frequent cause of critical illness after burn, blast, and prolonged immobility. No specific treatment exists, and supportive treatment is highly burdensome, contributing to deaths in austere environments, especially disasters and conflicts. Therefore, there is urgent need for specific treatment which reduces burden of care. They investigate whether cilastatin sodium, has efficacy as a crush syndrome treatment in a highly translational large animal trauma model.

The manuscript presents solid and relevant data on the use of sodium cilastatin to mitigate crush syndrome in a porcine model. The methodology is robust, the animal model has high translational validity, and the results are clinically significant. Renal function measurement and physiological and biochemical evaluation are accurate. The discussion is well contextualized within the field of crush syndrome research. The work has the potential to lay the groundwork for human clinical trials and stands out for its focus on an underexplored clinical issue in disaster scenarios.

Reviewer's Conclusion:

After carefully evaluating the authors' responses and the revisions undertaken to address the requested points, I consider that the manuscript now meets the standards required for publication. The authors have provided clear and well-referenced explanations, and the necessary modifications have been appropriately implemented. I would like to commend the authors for the high quality of their work and note that the study holds substantial promise for translational relevance to clinical practice. I sincerely acknowledge and appreciate the significant effort invested by the authors.

>>Thank you.

Reviewer #2 (Remarks to the Author):

This revised version of the article is more refined according to my comments . So I have no

further comments

>>Thank you.

Reviewer #3 (Remarks to the Author):

I greatly appreciate the authors' thoughtful consideration of my previous concerns and their thorough revision of the manuscript. I now have only a few minor comments:

1. Looking through the revised data (Figure 5A-C and 6C), I noticed that the effectiveness of cilastatin was limited. It did not consistently show a significant benefit across multiple timepoints. Therefore, I would recommend they avoid overpromising and tone down their claim.

>>We agree with the reviewer on the importance of avoiding overpromising, and we have changed the word “ameliorate” in the first sentence of discussion (and elsewhere) to the less emphatic “reduce”. We changed the penultimate section of discussion from:

Despite these limitations, this work provides strong evidence of the efficacy of cilastatin sodium to ameliorate crush syndrome, in a highly translational model.

to:

Despite these limitations, this work provides strong evidence of the efficacy of cilastatin sodium to **reduce kidney injury and the need for hyperkalemia treatment** due to crush syndrome, in a highly translational model.

Respectfully, we point out the primary outcome was successfully tested (48h GFR) and the corroborative outcomes (BUN and creatinine) are nonspecific (which is the reason the primary outcome was GFR). Secondary outcomes of clinical importance, including need for hyperkalemia intervention and tubular damage score are greatly improved by cilastatin. Nonetheless, these are secondary outcomes.

2. Generally speaking, if data are distributed normally, mean \pm SEM may be fine (mean \pm SD is more appropriate). If data are distributed nonnormally, they should show median (interquartile range) in their bar graphs (or box plots).

>>we now report nonnormally distributed data as median(25th percentile, 75th percentile) in the text and figures. Error bars represent the IQR in the affected figures, discussed in the prior round of review.

3. Please show scale bars in their histological images or specify the magnification in the figure legends.

>>We added scale bars to the histological images without them (in figure 3 and figure 5, and revised the figure captions to reflect the addition of scale bars.